# Suffering from chronic tinnitus, chronic neck pain, or both: Does it impact the presence of signs and symptoms of central sensitization?

**Kayleigh De Meulemeester**[1,2]*, **Mira Meeus**[1,2,3], **Robby De Pauw**[1,4], **Barbara Cagnie**[1], **Hannah Keppler**[5,6], **Dorine Lenoir**[1,2]

**1** Spine, Head and Pain Research Unit Ghent, Department of Rehabilitation Sciences, Faculty of Medicine and Health Sciences, Ghent University, Ghent, Belgium, **2** Pain in Motion International Research Group, The Netherlands, **3** MOVANT Research Group, Department of Rehabilitation Sciences and Physiotherapy, Faculty of Medicine and Health Sciences, University of Antwerp, Antwerp, Belgium, **4** Lifestyle and Chronic Diseases, Department of Epidemiology and Public Health, Sciensano, Belgium, **5** Audiology Research Group, Department of Rehabilitation Sciences, Faculty of Medicine and Health Sciences, Ghent University, Ghent, Belgium, **6** Department of Oto-Rhino-Laryngology, Ghent University Hospital, Ghent, Belgium

* Kayleigh.demeulemeester@ugent.be

**Data Availability Statement:** The data files will be made publicly available upon publication at www.clinicaltrials.gov (NCT05186259).

## Abstract

Chronic subjective tinnitus is a prevalent symptom, which has many similarities with chronic pain. Central sensitization is considered as a possible underlying mechanism of both symptoms. Central sensitization has already been investigated in chronic pain populations but not in patients with chronic subjective tinnitus. Therefore, the main objective of this cross-sectional study was to compare signs and symptoms, indicative for central sensitization, in tinnitus patients with and without chronic idiopathic neck pain, patients with chronic idiopathic neck pain only, and healthy controls. Also, differences in psychological and lifestyle factors, possibly influencing the association between central sensitization and tinnitus, were examined as well as correlations between signs and symptoms of central sensitization, and tinnitus, pain, psychological and lifestyle factors. Differences in signs and symptoms of central sensitization were examined using the self-report Central Sensitization Inventory and QST protocol (local and distant mechanical and heat hyperalgesia, conditioned pain modulation). Tinnitus, pain, psychological and lifestyle factors were evaluated using self-report questionnaires. Symptoms of central sensitization and local mechanical hyperalgesia were significantly more present in both tinnitus groups, compared to healthy controls, but were most extensive in the group with chronic tinnitus +chronic idiopathic neck pain. Distant mechanical hyperalgesia, indicative for central sensitization, was only observed in the group with both chronic tinnitus+chronic idiopathic neck pain. This group also displayed a significantly higher psychological burden and poorer sleep than patients with chronic tinnitus only and healthy controls. Signs and symptoms of central sensitization were also shown to be associated with tinnitus impact, pain-related disability, psychological burden and sleep disturbances. This study shows preliminary evidence for the presence of central sensitization in patients with chronic tinnitus+chronic idiopathic neck pain. This could be explained by the higher perceived tinnitus impact, psychological burden and sleep problems in this group.

**Funding:** The authors received no funding supporting this study.

**Competing interests:** The authors have declared that no competing interests exist.

**Trial registration:** This study is registered as NCT05186259 (www.clinicaltrials.gov).

## Introduction

Tinnitus is a frequent symptom with an important personal [1] and societal impact [2, 3]. The global prevalence of tinnitus ranges from 10 to 24% of the population and 10% of the population (pooled global prevalence) reports chronic tinnitus (present for more than three months) [4]. The most common type is subjective tinnitus (95% of all patients) and is defined as the conscious awareness of a tonal or composite noise for which there is no identifiable corresponding external acoustic source [5]. Tinnitus has been shown to often co-occur with (chronic) pain of which temporomandibular joint (TMJ) pain [6–8], headache [9–12] and neck pain [7, 12] are most reported. Remarkably, several similarities between chronic tinnitus and chronic pain have been identified [13–16], which can be divided into four clusters. First, both symptoms are frequently accompanied by hypersensitivity to sensory stimuli [15]. Patients with tinnitus often experience hyperacusis [17, 18], which can be accompanied by sound-induced pain at lower intensities when compared to healthy controls (HCs). Patients with chronic pain often display increased pain sensitivity for non-noxious stimuli (allodynia) and for noxious stimuli (hyperalgesia) [19]. Second, both in tinnitus and chronic pain, involvement of the central nervous system has been reported. Several animal studies and neuroimaging studies in humans, showed evidence for functional and structural brain alterations in patients with tinnitus [20–24] and chronic pain [25]. In addition, increasing evidence is found for the involvement of higher cognitive and affective brain areas in the pathophysiology of both conditions [16]. Third, both tinnitus [26–30] and chronic pain [31–34] are shown to be associated with psychological factors such as anxiety, fear, depression, stress and catastrophizing. In addition, the association between tinnitus-related distress and pain is shown to be mediated by psychological factors [35]. Lastly, several lifestyle factors such as sleep impairment and physical activity level are shown to be associated with both tinnitus [36–38] and chronic pain [39]. Given the similarities and associations between tinnitus and chronic pain, it is frequently hypothesized that these conditions share common underlying pathophysiological mechanisms [13–16, 40]. Since hypersensitivity to sensory stimuli [19, 41, 42] and involvement of the central nervous system [43, 44] are typical features of central sensitization (CS), this could be a possible candidate for a linking phenomenon between chronic pain and tinnitus. Central sensitization is defined as "an amplification of neural signaling within the central nervous system" [43] and is caused by maladaptive neuroplastic changes at different levels of the central nervous system [43, 44]. Central sensitization is already well documented in several chronic pain conditions [43, 45–48] but to the best of our knowledge, not yet investigated in patients with tinnitus. It can be hypothesized that signs and symptoms of CS are present in patients with chronic tinnitus, and more pronounced in patients with both chronic tinnitus and chronic pain. Therefore, the first aim of the present study is to compare signs and symptoms of CS between chronic tinnitus patients with and without chronic idiopathic neck pain (CINP), patients with CINP only and healthy controls. Since psychological and lifestyle factors are shown to be associated with tinnitus [36–38], chronic pain [39] and CS [49–51], and thus can influence the relationship between tinnitus and CS, the second aim is to compare psychological and lifestyle factors between all groups and a third aim is to explore how these signs and symptoms of CS are associated with psychological and lifestyle factors, as well as with factors related to tinnitus and pain. It can be expected that more deterioration in self-reported psychological and lifestyle measures is present in patients with chronic tinnitus or CINP, when

compared to HCs and are most prominent in patients with both chronic tinnitus and CINP. It can also be hypothesized that more extensive signs and symptoms of CS are associated with more deterioration in psychological, lifestyle, tinnitus and pain factors.

## Materials and methods

### Study design and setting

This cross-sectional study is part of a larger study, registered as NCT05186259 (www. clinicaltrials.gov). In this paper, only the primary research aim will be covered and thus not all outcome measures and patients, as mentioned in the clinical trial registration, were included for data analysis. These data will be included in a follow-up paper, covering the second research question, as mentioned in the clinical trial registration. Chronic tinnitus patients (CTIN), chronic tinnitus patients with CINP (CTIN+CINP), patients with CINP only, and HCs were evaluated for signs of CS, using a Quantitative Sensory Testing (QST) protocol at the research laboratories of the department of Rehabilitation Sciences at Ghent University (Ghent, Belgium). All participants were also asked to fill out self-report questionnaires regarding self-reported symptoms of CS, tinnitus, pain, psychological and lifestyle factors (online survey). The study was approved by the Ethics committee of Ghent University Hospital on 24/03/2020 (approval number: BC-07036).

### Participants

Subjects were recruited by means of flyers and posters, which were spread in public places, general practitioner, dentist and physiotherapists practices, Ghent University Hospital (Ear, Nose and Throat department) and other Flemish hospitals, pharmacies, music schools, large companies, and social media. The inclusion and exclusion criteria for all participants are shown in Tables 1 and 2. Patients with CTIN+CINP or CINP were required to report a mean pain intensity of at least 3/10 on a numeric rating scale (NRS), since this is defined as the cut-off for clinically relevant pain [52].

**Table 1. General inclusion and exclusion criteria (applicable for all groups).**

| Inclusion criteria | Exclusion criteria |
|---|---|
| Aged between 18–65 years | Vertigo (Menière's disease, BPPV, etc) |
| Speaking and understanding Dutch fluently | Deafness |
| | Subjects with prior otologic surgery (for example stapedotomy), active outer or middle ear pathology |
| | Wearing a hearing aid device, implant, noise generators or receiving neuromodulation therapy |
| | History of head, neck or shoulder trauma or surgery (< 5 years, or remaining complaints) or a history of whiplash trauma |
| | Diagnosis of fibromyalgia/chronic fatigue syndrome |
| | Life threatening, metabolic, cardiovascular, neurologic, systemic diseases or intracranial pathologies |
| | Major depression or psychiatric illness (diagnosed by a psychiatrist and being in medicamental or psychiatric treatment) |
| | Pregnancy or given birth in the preceding year |
| | Dyslexia, dyscalculia, AD(H)D, language/communication disorder |
| | Taking medication that has a negative influence on cognition |

**Table 2. Group specific inclusion and exclusion criteria.**

| Inclusion criteria | Exclusion criteria |
|---|---|
| **Patients with CTIN** | |
| Chronic subjective tinnitus ($\geq$ 3 months during most of the days (4 or more)and for more than 5 minutes/day) (5) | Objective tinnitus |
| | Subjective tinnitus caused by clear causes such as tumor, trauma, vascular dysfunction, neurological disorder |
| | Chronic musculoskeletal pain ($\geq$ 3 months with mean pain intensity $\geq$ 3/10 in the preceding month) |
| | Pain in any region with a pain intensity > 2/10 on the day of testing |
| **Patients with CTIN and CINP** | |
| Chronic subjective tinnitus ($\geq$ 3 months during most of the days (4 or more) and for more than 5 minutes/day) | Objective tinnitus |
| Chronic idiopathic neck pain ($\geq$ 3 months with a mean pain intensity $\geq$ 3/10 in the preceding month) | Subjective tinnitus caused by clear causes such as tumor, trauma, vascular dysfunction, neurological disorder |
| Reporting at least mild disability ($\geq$5/42) on the Neck Disability Index (53) | Specific causes of neck pain, such as cervical hernias with clinical symptoms |
| **Patients with CINP** | |
| Chronic idiopathic neck pain ($\geq$ 3 months with a mean pain intensity $\geq$ 3/10 in the preceding month) | Any form of tinnitus or hyperacusis |
| Reporting at least mild disability ($\geq$5/42) on the Neck Disability Index (53) | Specific causes of neck pain, such as cervical hernias with clinical symptoms |
| | Diagnosis of any TMD, according to the Research Diagnostic Criteria for TMD (RDC/TMD); or concomitant diagnosis of primary headache |
| **Healthy controls** | |
| | Any form of tinnitus or hyperacusis |
| | Experiencing any type of pain during at least 8 consecutive days with a mean pain intensity > 2/10 in the preceding year |
| | Pain in any region with a pain intensity > 2/10 on the day of testing |

## Procedure

Interested volunteers were asked to complete an online inclusion questionnaire to check for eligibility. After inclusion, participants were invited for a lab visit for experimental assessments (1 h 45 minutes). Prior, participants were asked to fill out an online baseline questionnaire battery (1 hour, fixed order) to acquire information on demographics, in-and exclusion criteria, tinnitus- and pain-related characteristics, psychological and lifestyle factors. Participants were also instructed, and compliance was registered at the start, to refrain from vigorous physical activities (> 10 metabolic equivalents, where 1 metabolic equivalent is defined as the amount of oxygen consumed while sitting at rest [53]), alcohol, nicotine and caffeine consumption (24 hours before lab visit) and pain medication (48 hours before lab visit). In addition, participants provided written informed consent and were asked to report current pain complaints on a pain drawing and NRS (0–10). The experimental assessments consisted of signs of CS by means of QST, using mechanical and heat stimuli during static and dynamic measurements. All measurements were performed by four physiotherapists who were trained together to follow the same standardized test protocol.

## Quantitative Sensory Testing: Static measures

Pressure pain thresholds (PPTs) and cutaneous heat pain thresholds (HPTs) were measured at the neck [54, 55] and trigeminal region (Masseter muscle [56, 57] and Trigeminal nerve [58]) to evaluate local hyperalgesia, and at the lateral elbow [59] and Tibialis Anterior muscle [54, 60], to evaluate distant hyperalgesia. Distant hyperalgesia at the Tibialis Anterior muscle was set as primary outcome measure since distant hyperalgesia is commonly regarded as an indicator of CS [61, 62]. The measures were performed in a fixed order at standardized marked locations (S1 Table) on the most painful side in the groups with CINP and at the dominant side in the groups without CINP. PPTs and HPTs at these locations have been used in several studies in neck pain [48, 54, 55, 60, 63–66], TMJ pain [67, 68], headache [69–72], and healthy participants [59, 73]. PPT and HPT measurements were first demonstrated at the non-tested palmar face of the hand to familiarize the patient with the procedure.

**Pressure pain thresholds (local and distant mechanical hyperalgesia).** PPTs were measured unilaterally with a digital pressure algometer (FDX TM, Wagner Instruments, Greenwich, Connecticut). The ascending method of limits was applied, and pressure was gradually increased at a rate of 1 kgf/s. The participants were instructed to say "yes" when the sensation changed from pressure to pain (PPT) and when the pain reached an intensity of (6/10) (6PPT), which terminated the pressure stimulus. The (6)PPTs at each site were determined as the mean of 2 consecutive (with a 30 seconds rest interval in between) measurements (75). Moderate to excellent intra-rater and inter-rater reliability has been shown for PPTs in neck pain patients [60, 74] and healthy participants [60, 75].

**Cutaneous heat pain thresholds (local and distant heat hyperalgesia).** Cutaneous HPTs were evaluated by means of the ascending method of limits, using the Contact Heat-Evoked Potentials (CHEPS) Pathway device (Medoc). An Advanced Thermal Stimulator (ATS) thermode was applied directly to the skin and held in place manually at the predefined marks. The baseline temperature was set at 32˚C and increased at a rate of 1˚C/s up to a maximum temperature of 51˚C. Participants held a dual response button, and were asked to press the first button when the heat sensation changed into a pain sensation (HPT) and to press a second button when the pain intensity reached 6/10 (6HPT). The temperature of the HPT and 6HPT was recorded and turned back to baseline temperature when the 6HPT threshold or the maximum temperature of 51˚C was reached. The (6)HPTs at each site were determined as the mean of three consecutive (with a 15 seconds rest interval in between) measurements. In case the HPT or 6HPT was not reached before the maximum temperature of 51˚C, the threshold was recorded as 51˚C for that trial [54]. The assessment of HPTs shows fair to good reliability in pain patients [76] and moderate to excellent intra- and inter-rater reliability in healthy controls [75].

## Quantitative Sensory Testing: Dynamic measures

**Conditioned pain modulation: Endogenous pain inhibition.** The efficacy of endogenous pain inhibition was assessed using a conditioned pain modulation (CPM) paradigm. In this psychophysical paradigm, a noxious stimulus is used as a conditioning stimulus to reduce pain perception from another noxious test stimulus, reflecting the "pain inhibits pain" mechanism [77, 78]. The conditioning stimulus for eliciting CPM consisted of immersion of the non-dominant or non-painful hand up to the wrist into a bath (VersaCool™, Thermo Fisher Scientific Inc., Waltham, Massachusetts USA) with circulating hot water maintained at 45.5˚C for one minute, which is shown to have fair to excellent reliability [79] elicits a robust CPM effect, without potential ceiling or floor effects [80]. According to the recommendations of Yarnitsky et al. (2015) a sequential procedure was used with two modalities for test stimuli

[81], being first mechanical pain stimulation using pressure algometry and second heat pain stimulation, both at the lateral elbow of the dominant or painful side. (6)PPTs and (6)HPTs were determined as described in the static measures section. The use of a test stimulus eliciting a pain intensity of 6/10 is in accordance with previous studies [82–84]. The test stimulus was applied before and immediately after hot water immersion. The use of PPTs has been shown to be a valid [59] and reliable [79, 85] test stimulus for CPM assessment whereas the use of HPTs is shown to be an effective test stimulus for CPM [85] but has lower reliability [79, 85]. Pain intensity for the hot water conditioning stimulus was evaluated using an NRS from 0 to 10, with 0 referring to "no pain" and 10 to "maximal pain" felt.

For analyses of CPM efficacy, both absolute CPM effect and relative CPM effect were calculated [81]. To calculate the absolute CPM effect, the mean (6)PPT/(6)HPT measured before the hot water immersion (preCPM) was subtracted from the mean measured after the hot water immersion (postCPM). Hence, a lower CPM value reflected less efficient endogenous pain inhibition. The relative CPM effect (percent change) was calculated with the following formula: [(absolute CPM effect)/preCPM]*100 [86]. Based on the calculation of the relative CPM effect, a positive value represented a responder and a negative value a non-responder.

**Temporal summation: Endogenous pain facilitation.** Temporal summation of heat pain was assessed as a measure of endogenous pain facilitation but due to technical errors during the testing, this outcome was not used for further analysis.

### Self-report questionnaires

**Self-reported tinnitus-related and pain-related measures: Tinnitus Sample Case History Questionnaire, Tinnitus Functional Index, Hyperacusis Questionnaire and Neck Disability Index.** The Dutch validated version of the Tinnitus Sample Case History Questionnaire (TSCHQ) was used for the standardized collection of information regarding the tinnitus history and characteristics, and other symptoms such as neck pain or headache [87].

The Dutch validated version of the Tinnitus Functional Index (TFI) was used to evaluate tinnitus impact. The TFI is a 25-item self-report questionnaire with a total score of 100, which consists of eight different subscales (intrusiveness, cognition, sleep, sense of control, relaxation, emotional, auditory and quality of life subscales) [88]. Tinnitus impact, based on the TFI, can be categorized as mild tinnitus ($\leq$ 25), significant tinnitus (26–50), and severe tinnitus (>50) [89]. The Dutch version of the TFI shows good reliability and internal consistency (Cronbach's alpha = 0.96) [88].

The Dutch version of the Hyperacusis Questionnaire (HQ) was used for the quantification and characterization of hyperacusis. The HQ consists of 14 items with a total score of 42, a score greater than 28 is considered to represent auditory hypersensitivity or hyperacusis [90] The Dutch version of the Hyperacusis Questionnaire shows good internal consistency (Cronbach's alpha value of 0.85) [91].

The Dutch version of the Neck Disability Index (NDI) was used to evaluate the level of self-reported pain-related disability [92, 93]. The NDI consists of 10 items and has a total score of 50. Higher NDI scores reflect higher levels of neck pain-related disability. A score between 0 and 4 reflects no disability, a score between 5 and 14 indicates mild disability, between 15 and 24 moderate disability, between 25 and 34 severe disability and > 35 is considered as complete disability [93]. The Dutch version of the NDI has been shown to be reliable and valid, and has good internal consistency (Cronbach's alpha: 0.87) in patients with chronic neck pain [94, 95].

**Self-reported symptoms of CS: Central Sensitization Inventory.** Self-reported symptoms indicative of CS were evaluated using the Dutch version of the Central Sensitization Inventory (CSI) [96]. A cutoff of 40 out of 100 is used to determine the presence of self-

reported signs of CS (the higher the score, the higher the severity). The Dutch CSI has shown excellent test-retest reliability, good discriminative strength and internal consistency (Cronbach's alpha:0.60–0.89) in patients with chronic pain [97].

**Self-reported psychological factors: Connor-Davidson Resilience Scale, Depression, Anxiety and Stress Scale-21, Beck Depression Inventory, Big Five Index -2, Pain Catastrophizing Scale.** The Dutch version of the Connor-Davidson Resilience Scale 25 (CD-RISC 25) was used to assess resilience, which is a measure of stress coping ability [98, 99]. The CD-RISC 25 consists of 25 items with a total score of 100 and a higher score reflecting greater resilience. The CD-RISC 25 has shown good psychometric properties and can distinguish between people with greater and lesser resilience [98, 100], also in patients with chronic pain [101]. The Dutch version has shown good internal consistency (Cronbach's alpha: 0.90) [102].

Three negative emotional dimensions: 'depression', 'anxiety' and 'stress were evaluated using the self-report Depression Anxiety and Stress Scale 21 (DASS21), which is a short version of the DASS [103, 104]. The questionnaire consists of 21 questions of which the score was multiplied by 2 so a subscale score of maximal 42 and a total score of maximal 126 can be reached. Higher scores indicate more severe negative emotional status. Five levels of severity can be distinguished for each subscale (normal, mild, moderate, severe, extremely severe). The Dutch translation of the DASS21 has shown sufficient to good validity, internal consistency (Cronbach's alpha: 0.85–0.94) and test retest reliability [105].

The Dutch version of the Beck Depression Inventory (BDI) was used for the assessment of depression [106, 107]. The BDI has a total score of 63 and higher scores reflect more severe depression. The cut-off score for depression in non-psychiatric populations is considered to be ≥13/63 [108]. The BDI has shown to be a valid and reliable tool to evaluate depressive symptoms in patients with chronic pain [109]. It is also commonly used to assess depression in patients with tinnitus [30].

The Big Five Index 2 (BFI-2) was used to quantify five traits of personality; agreeableness, conscientiousness, extraversion, neuroticism, and openness [110]. The BFI-2 consists of 15 facets, describing different features of each trait [111]. The BFI-2 has shown good validity and reliability [110].

Neck pain catastrophizing was assessed using the Dutch Pain Catastrophizing Scale (PCS), which is a self-report questionnaire to evaluate the presence of catastrophic thoughts and feelings towards pain [112, 113]. The PCS consists of 13 items and has a maximum score of 52. Higher scores indicate higher levels of pain catastrophizing. The PCS is found to be valid and to have acceptable to good internal consistency (Cronbach's alpha: 0.75–0.93) [112]. The Dutch version of the PCS exhibits moderate test-retest reliability [114].

**Self-reported lifestyle factors: Baecke Physical Activity Questionnaire, Pittsburgh Sleep Quality Index, Insomnia Severity Index.** Self-reported physical activity levels were evaluated using the Baecke Physical Activity Questionnaire [115]. This questionnaire consists of 16 items assessing three different domains of physical activity: work, sports and leisure time. The total score varies between 3 and 15 with a higher score reflecting a greater level of physical activity. The Baecke questionnaire has shown moderate to excellent reliability but only fair correlations with accelerometer data in patients with chronic low back pain [116].

The Pittsburgh Sleep Quality Index (PSQI) was used to evaluate self-perceived overall sleep quality in 7 domains: subjective sleep quality, sleep latency, sleep duration, sleep efficiency, sleep disturbance, sleep medication, and daytime dysfunction over the previous month [117]. The maximal score is 21, and a global score of 5 or higher indicates clinically significant sleep problems [117, 118]. The PSQI shows good validity and sufficient to good internal consistency (0.70–0.83) in non-clinical and clinical populations [119], and in patients with chronic pain [120].

The Insomnia Severity Index (ISI) was used to assess the patient's perception of insomnia severity [121]. The ISI consists of seven items assessing the severity of sleep onset and sleep maintenance difficulties (both nocturnal and early morning awakenings), satisfaction with current sleep pattern, interference with daily functioning, notice ability of impairment attributed to the sleep problem and degree of distress or concern caused by the sleep problem. A score between 0 and 7 indicates no clinically significant insomnia, between 8 and 14 is interpreted as subthreshold insomnia, 15–21 as clinical insomnia with moderate severity, and 22–28 as severe clinical insomnia. Total score ranges from 0 to 28 (with higher scores indicative of more severe insomnia). The ISI has shown good validity and test-retest reliability [121, 122]. Both PSQI and ISI are shown to have good validity and reliability to assess sleep dysfunctions in patients with chronic pain [123] and have also been used in tinnitus research [36, 37].

## Data analysis

All statistical analyses were performed with IBM SPSS Statistics 25.0 (IBM SPSS, Armonk, New York). For demographic data, the normality of continuous variables was evaluated using the Shapiro-Wilk test and by visual evaluation of histograms, QQplots and boxplots. The equality of variance was examined using the Levene's test. Continuous data were analyzed using parametric tests (Unpaired student's T test for 2-group comparisons and One-way ANOVA for 4-group comparisons with post hoc Bonferroni ($P<0.01$)) in case of normal distribution and equality of variance. In case these assumptions were not met, non-parametric tests were applied (Mann Whitney-U test for 2-group comparisons and Kruskal-Wallis test for 4-group comparisons with post hoc Mann-Whitney U test). For the categorical data, Chi squared, or Fisher's exact tests were used with post hoc Bonferroni correction ($P<0.01$) if necessary.

For the QST measures and self-report questionnaires, univariate or multivariate analyses of covariance ((M)ANCOVA) were performed to determine group differences for all measures. Dependent variables which showed significant Pearson correlations ($P<0.05$) were clustered into one model. Age and sex are shown to possibly influence QST measures [48, 124, 125], and were therefore included as covariates in the QST models. Assumptions for normal distribution of residuals were checked by visual graph inspection and homogeneity of variance was checked by visual graph inspection and Levene's test. In case a significant group effect was found ($P<0.05$), post hoc pairwise comparisons were performed. To correct for multiple comparisons (in case of five comparisons: CTIN vs HC, CTIN+CINP vs HC, CINP vs HC, CTIN vs CTIN+CINP, CTIN+CINP vs CINP), only differences with a significance below $\alpha < 0.01$ (Bonferroni correction: 0.05/5) were assumed to be significant. In case only two groups were compared (for TFI, HQ, NDI and PCS), the significance level was set at $P<0.05$. Effect sizes (ES; Cohen $d$) and relevant confidence intervals between each group for adjusted means were calculated using a standard formula (124). Effect sizes from 0.2 to 0.49 were considered small, 0.5 to 0.79 considered moderate, and 0.8 and above as large [126].

Pearson correlation analyses were performed between QST measures and the self-report measures (total scores of questionnaires). To correct for multiple comparisons, we deemed only Pearson correlations $P<0.013$ (2-tailed) to be significant for correlation analysis in the whole group (Bonferroni correction: 0.05/4, since associations were analyzed between QST measures and four clusters of questionnaires, including tinnitus (TFI, HQ), pain (NDI), psychological (CDRISC-25, DASS21, BDI, BFI-2) and lifestyle factors (PSQI, ISI, Baecke). For correlation analysis of the chronic tinnitus group (TFI and HQ) and of the CINP group (NDI and PCS), significance was set at 0.05.

An a priori sample size calculation was performed in G*Power (version 3.1.9.2) for the primary outcome measure distant PPT at the M. Tibialis Anterior. Because of the absence of data for tinnitus patients, the Cohen d effect size was calculated based on the PPT data for the Tibialis Anterior muscle of a previous study [127] in which a patient group with chronic neck pain was compared to a healthy control group. Based on this a priori sample size calculation with a Cohen d effect size of 1.32, a significance level of 0.01 (Bonferroni correction (0.05/5)), and a power of 0.80 (ANOVA: Fixed effects, omnibus, one-way), a total sample size of at least 66 participants was required, with 17 participants per group.

## Results

### Population characteristics

Forty-seven patients with CTIN, 19 patients with CTIN+CINP, 29 patients with CINP and 40 HCs were found to be eligible and participated in the study. A detailed overview of patient enrollment and data collection during each phase of the study can be found in Fig 1. Additional missing data for a specific variable are listed in the tables or figures. The QST measurements were performed on the right body side in 83% of the CTIN, 65% of the CTIN + CINP, 67% of the CINP patients and 75% of the HCs.

### Demographic characteristics, self-reported tinnitus and pain measures

The demographic characteristics, self-reported tinnitus and pain measures are shown in Table 3. Sex was shown to be significantly different between the groups and was therefore included as a covariate in all analyses. Several participants indicated to have not refrained from nicotine, alcohol, caffeine and vigorous activity in the 24 hours before QST measurements. However, only nicotine (P = 0.032) and caffeine (P<0.001) consumption was shown to be significantly different between groups. These variables were therefore also included as covariates in the (M)ANCOVA analyses for QST. During the 48 hours prior to the QST measurements, one CTIN patient reported the use of paracetamol and three CINP patients reported the use of NSAIDs. None of the CTIN+CINP patients or HCs reported the use of analgesic medication. A univariate analysis of covariance showed a significantly higher TFI score in the CTIN+CINP group, compared to CTIN (Table 3, Fig 3A). Univariate analyses of covariance displayed no significant group differences for HQ or NDI (Table 3).

### Between-group differences in Quantitative Sensory Testing measures

**Static measures: Pressure pain thresholds and heat pain thresholds.** A multivariate analysis of covariance revealed significant group differences for PPT at the Trigeminal nerve, Masseter and Tibialis Anterior muscle and for 6PPT at all locations (Table 4, Fig 2A and 2B). When compared to HC, the CTIN group only displayed a significantly lower 6PPT at the Trigeminal nerve whereas the CTIN+CINP group had significantly lower (6)PPTs at all these sites. The CINP group had significantly lower 6PPTs at the Masseter muscle and Trigeminal nerve. The CTIN+CINP group showed significantly lower thresholds at the Trigeminal nerve (PPT) and Tibialis Anterior muscle (PPT and 6PPT), compared to the CTIN group. No significant differences were found between CTIN+CINP and CINP. A multivariate analysis of covariance revealed no significant group differences for HPT or 6HPT at all locations (Table 4). For several (6)HPT measurements, the 51˚ C threshold was reached before reaching the (6)HPT (Table 4).

**Dynamic measures: Conditioned pain modulation.** A multivariate analysis of covariance revealed no significant group differences for absolute or relative CPM effect at all

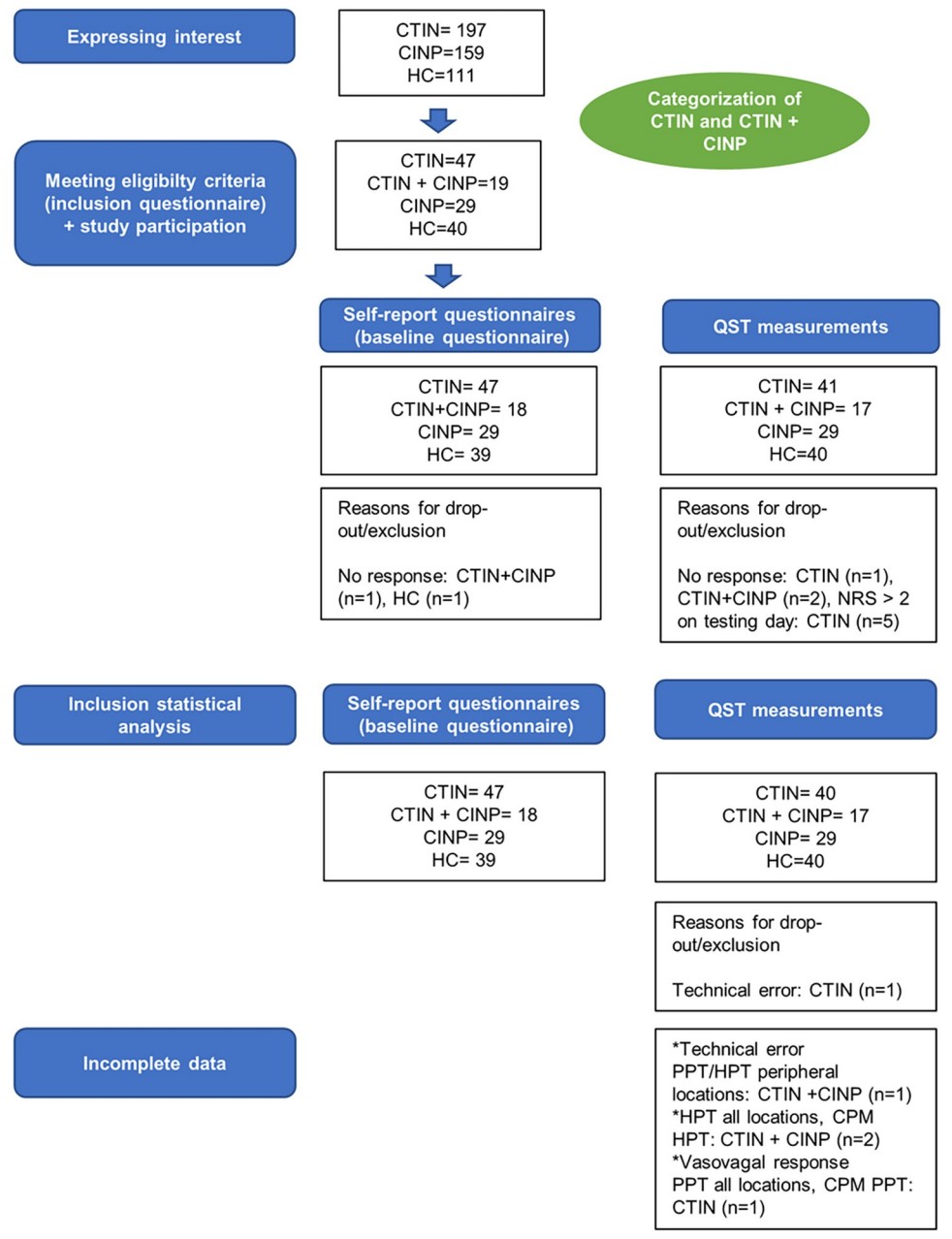

**Fig 1. Flowchart of participant enrollment in each study phase.** CTIN: chronic tinnitus, CINP: chronic idiopathic neck pain, HC: healthy controls, CTIN+CINP: chronic tinnitus and idiopathic neck pain, QST: quantitative sensory testing, PPT: pressure pain threshold, HPT: heat pain threshold.

locations (Table 4). As shown in Table 6, the majority of participants displayed a CPM effect and were classified as responders.

## Between-group differences in self-report questionnaires

**Self-reported symptoms of CS.** A univariate analysis of covariance showed a significant group difference for the CSI (Table 5, Fig 3A). All patient groups showed significantly higher

**Table 3. Demographic, tinnitus and pain characteristics.**

*Demographics (inclusion and baseline questionnaire)*

| | Group | Mean Frequencies (n (%)) | Median | Range (min-max) | SD | IQR | P-value |
|---|---|---|---|---|---|---|---|
| Age (y) [a] | HC | 36.42 | 30.00 | 19.0–63.0 | 14.148 | 25.0 | 0.873 |
| | CTIN | 37.85 | 37.00 | 21.0–60.0 | 13.678 | 26.0 | |
| | CTIN+CINP | 39.32 | 35.00 | 22.0–62.0 | 13.237 | 25.0 | |
| | CINP | 37.76 | 33.00 | 18.0–63.0 | 15.408 | 29.0 | |
| Body mass index (kg/m²) [a] | HC | 22.87 | 22.80 | 19.2–26.1 | 1.803 | 2.8 | 0.950 |
| | CTIN | 23.35 | 23.36 | 18.3–30.1 | 2.904 | 3.5 | |
| | CTIN+CINP | 24.24 | 22.90 | 19.6–32.1 | 4.096 | 7.1 | |
| | CINP | 23.36 | 23.83 | 18.7–32.3 | 3.173 | 4.8 | |

**Sex [b]** — P-value <0.001

| | Group | Men | Women |
|---|---|---|---|
| | HC | 16 (40.0) | 24 (60.0) |
| | CTIN | 34 (72.3) | 13 (27.7) |
| | CTIN+CINP | 6 (31.6) | 13 (68.4) |
| | CINP | 7 (24.1) | 22 (75.9) |

**Hand preference [c]** — P-value 0.125

| | Group | Left | Right | Ambidexter |
|---|---|---|---|---|
| | HC | 10 (25.0) | 30 (75.0) | 0 (0) |
| | CTIN | 7 (14.9) | 40 (85.1) | 0 (0) |
| | CTIN+CINP | 1 (5.3) | 18 (94.7) | 0 (0) |
| | CINP | 2 (6.9) | 26 (22.8) | 1 (3.4) |

**Education level [c]** — P-value 0.228

| | Group | No degree | Lower secondary | Higher secondary | Higher education |
|---|---|---|---|---|---|
| | HC[g] | 1 (3.6) | 1 (3.6) | 2 (7.1) | 24 (85.7) |
| | CTIN | 0 (0) | 9 (19.1) | 4 (8.5) | 34 (72.3) |
| | CTIN+CINP | 1 (5.3) | 5 (26.3) | 1 (5.3) | 12 (63.2) |
| | CINP | 2 (6.9) | 5 (17.2) | 4 (13.8) | 18 (62.1) |

**Work status [c]** — P-value 0.908

| | Group | Student | Employed | Unable | Retired | Unemployed | Other |
|---|---|---|---|---|---|---|---|
| | HC | 8 (20.5) | 27 (69.2) | 0 (0) | 2 (5.1) | 1 (2.6) | 1 (2.6) |
| | CTIN | 11 (23.4) | 34 (72.3) | 0 (0) | 0 (0) | 2 (4.3) | 0 (0) |
| | CTIN+CINP | 2 (11.1) | 14 (77.78) | 1 (5.6) | 1 (5.6) | 0 (0) | 0 (0) |
| | CINP | 6 (20.7) | 20 (69.0) | 0 (0) | 3 (10.3) | 0 (0) | 0 (0) |

| | Group | Mean Frequencies (n (%)) | Median | Range (min-max) | SD | IQR | P-value |
|---|---|---|---|---|---|---|---|
| Duration tinnitus (months) [d] | CTIN | 112.58 | 90.00 | 3.0–396.0 | 85.923 | 120.0 | 0.908 |
| | CTIN+CINP | 126.70 | 96.00 | 13.2–480.0 | 114.945 | 132.0 | |
| Tinnitus loudness (VAS/0-100) [e] | CTIN | 39.47 | 30.00 | 3.0–85.0 | 24.738 | 45.0 | 0.710 |
| | CTIN+CINP | 42.00 | 37.50 | 3.0–85.0 | 23.659 | 37.0 | |

**Tinnitus location [c]** — P-value 0.867

| | Group | Only left | Only right | Bilateral | Bilateral (left dominant) | Bilateral (right dominant) | Head |
|---|---|---|---|---|---|---|---|
| | CTIN | 3 (6.4) | 2 (4.3) | 19 (40.4) | 12 (25.5) | 8 (17.0) | 3 (6.4) |
| | CTIN+CINP | 3 (16.7) | 1 (5.6) | 6 (33.3) | 4 (22.2) | 3 (16.7) | 1 (5.6) |

**Tinnitus continuity [b]** — P-value 0.748

| | Group | Intermittent | Constant |
|---|---|---|---|
| | CTIN | 10 (21.3) | 37 (78.8) |
| | CTIN+CINP | 5 (26.3) | 14 (73.7) |

**Diagnosis of tinnitus by ENT [b]** — P-value 0.519

| | Group | Yes | No |
|---|---|---|---|
| | CTIN | 9 (19.1) | 38 (80.9) |
| | CTIN+CINP | 5 (26.3) | 14 (73.7) |

*Tinnitus questionnaires (baseline questionnaire)*

| | Group | Mean (SE) | MD (95%CI)[*] | SMD (95% CI)[*] | P-value |
|---|---|---|---|---|---|
| TFI (/100) [f] | CTIN | 17.02 (2.845) | -13.20 (-23.88;-2.52) | -3.99 (-5.07;-2.79) | **0.016** |
| | CTIN+CINP | 30.22 (4.319) | | | |
| HQ (/42) [f] | CTIN | 18.35 (1.205) | -1.54 (-6.06;2.98) | -1.10 (-3.04;0.88) | 0.499 |
| | CTIN+CINP | 19.89 (1.829) | | | |

| | Group | Mean Frequencies (n (%)) | Median | Range (min-max) | SD | IQR | P-value |
|---|---|---|---|---|---|---|---|

(*Continued*)

**Table 3.** (Continued)

| | Group | Mean Frequencies (n (%)) | | | | | Median | Range (min-max) | SD | IQR | P-value |
|---|---|---|---|---|---|---|---|---|---|---|---|
| Mean neck pain intensity preceding month (NRS/0-10) [d] | CINP | 5.60 | | | | | 6.0 | 3.0–8.0 | 1.472 | 3.0 | 0.154 |
| | CTIN+CINP | 4.92 | | | | | 5.0 | 3.0–8.5 | 1.782 | 4.0 | |
| Neck pain duration (months) [d] | CINP | 67.20 | | | | | 30.00 | 6.0–360.0 | 76.462 | 78.0 | 0.530 |
| | CTIN+CINP | 75.75 | | | | | 66.0 | 3.0–180.0 | 62.971 | 96.0 | |
| Days/week neck pain preceding month [d] | CINP | 5.17 | | | | | 5.00 | 1–7 | 1.794 | 6 | 0.821 |
| | CTIN+CINP | 4.94 | | | | | 5.50 | 2–7 | 2.127 | 4 | |
| (Dominant) painful side [b] | | *Left* | *Right* | *Bilateral* | | | | | | | 0.824 |
| | CINP | 7(57.1) | 12(66.7) | 10(31.9) | | | | | | | |
| | CTIN+CINP[$] | 6(35.3) | 6(35.3) | 5(29.41) | | | | | | | |
| Other chronic pain complaints | | *TMJ* | *Headache* | *Low back* | *Upper limb* | *Lower limb* | | | | | |
| | CTIN+CINP | 4(13.8) | 5(17.2) | 6(20.7) | 3(10.3) | 1(3.4) | | | | | |

| *Pain Questionnaires (baseline questionnaire)* | Group | Mean (SE) | Median | MD (95%CI)* | SMD (95% CI)* | P-value |
|---|---|---|---|---|---|---|
| NDI1 (/50) [f] | CINP | 11.07(0.745) | -0.23(-2.45;1.99) | | 0.28 (-0.50; 1.05) | 0.836 |
| | CTIN+CINP | 11.30(0.883) | | | | |

*Compliance with guidelines 24 to 48 hours before QST testing*

| | Group | Frequencies (n (%)) | | P-value |
|---|---|---|---|---|
| Nicotine use [c] | | *No* | *Yes* | 0.002 |
| | HC | 39(97.5) | 1(2.5) | |
| | CTIN | 40(100.0) | 0(0.0) | |
| | CTIN+CINP | 17(100.0) | 0(0.0) | |
| | CINP | 23(79.3) | 6(20.7) | |
| Alcohol use [c] | | *No* | *Yes* | 0.947 |
| | HC | 29(72.5) | 11(27.5) | |
| | CTIN | 31(77.5) | 9(22.5) | |
| | CTIN+CINP | 13(76.5) | 4(23.5) | |
| | CINP | 23(79.3) | 6(20.7) | |
| Caffeine use [b] | | *No* | *Yes* | <0.001 |
| | HC | 13(32.5) | 27(67.5) | |
| | CTIN | 30(75.0) | 10(25.0) | |
| | CTIN+CINP | 13(76.5) | 4(23.5) | |
| | CINP | 6(20.7) | 23(79.3) | |
| Vigorous physical activities [c] | | *No* | *Yes* | 0.373 |
| | HC | 39(97.5) | 1(2.5) | |
| | CTIN | 39(97.5) | 1(2.5) | |
| | CTIN+CINP | 17(100.0) | 0(0.0) | |
| | CINP | 26(89.7) | 3(10.3) | |
| Pain medication use [c] | | *No* | *Yes* | 0.174 |
| | HC | 40(100.0) | 0(0.0) | |
| | CTIN | 38(95.0) | 2(5.0) | |
| | CTIN+CINP | 16(94.1) | 1(5.9) | |
| | CINP[$] | 25(89.3) | 3(10.7) | |

[a]Kruskall Wallis Test, [b]Chi-Square test, [c]Fisher's Exact Test, [d]Mann-Whitney U test, [e]Unpaired Student's T test, [f]Univariate analysis of covariance including sex as covariate.

[$]Data on education are missing from 12 HC, data on dominant painful side are missing from 2 CTIN+CINP, data on medication use are missing from one CINP

*(Standardized) mean differences were calculated as CTIN- CTIN+CINP or CINP-CTIN+CINP

N: sample size, min: minimum, max: maximum, SD: standard deviation, IQR: interquartile range, y: years, HC: healthy controls, CTIN: patients with chronic tinnitus, CTIN+CINP: patients with chronic tinnitus and chronic idiopathic neck pain, CINP: patients with chronic idiopathic neck pain, VAS: Visual Analogue Scale, ENT: ear nose throat specialist, SE: Standard Error, MD: Mean Difference, CI: Confidence Interval, SMD: Standardized Mean Difference, TFI: Tinnitus Functional Index, HQ: Hyperacusis Questionnaire, NRS: numeric rating scale, NDI: Neck Disability Index

**Table 4. Adjusted means and standard errors, and results of multivariate analysis of covariance (MANCOVA) of Quantitative Sensory Testing measures.**

*Static measures: pressure pain thresholds and heat pain thresholds*

| | HC (A) | CTIN (B) | | | CTIN + CINP (C) | | | CINP (D) | | |
|---|---|---|---|---|---|---|---|---|---|---|
| | Mean (SE) | Mean (SE), *MD (95% CI)** | SMD (95% CI)* | P | Mean (SE), *MD (95% CI)** | SMD (95% CI)* | P | Mean (SE), *MD (95% CI)** | SMD(95% CI)* | P |
| **C5C6 PPT** | 5.58 (0.800) | 5.26(0.837) | | | 4.30(0.923) | | | 5.07(0.771) | | |
| **Vs HC** | | *0.33(-0.61;1.26)* | 0.39 (-0.20;0.97) | 0.489 | *1.28(0.13;2.44)* | 1.53 (0.66;2.35) | 0.030 | *0.51(-0.45;1.47)* | 0.65 (-0.01;1.28) | 0.295 |
| **Vs CTIN +CINP** | | *-0.96(-2.09;0.17)* | -1.11(-1.91;-0.29) | 0.096 | | | | *-0.77(-2.02;0.48)* | -0.93(-1.75;-0.07) | 0.223 |
| **C5C6 6PPT** | 8.71 (1.043) | 7.44(1.091) | | | 6.24(1.204) | | | 7.10(1.005) | | |
| **Vs HC** | | *1.27(0.05;2.49)* | 1.19 (0.55;1.81) | 0.041 | **2.47(0.96;3.98)** | **2.27 (1.29;3.16)** | **0.002** | 1.61(0.35;2.86) | 1.57(0.83;2.25) | 0.012 |
| **Vs CTIN +CINP** | | *-1.19(-2.67;0.28)* | -1.07(-1.86;-0.24) | 0.111 | | | | *-0.86(-2.49;0.77)* | -0.80 (-1.61;0.05) | 0.298 |
| **MAS PPT** | 2.94 (0.466) | 2.58(0.487) | | | 1.94(0.538) | | | 2.24(0.449) | | |
| **Vs HC** | | *0.36(-0.19;0.90)* | 0.76 (0.15;1.35) | 0.198 | **1.00(0.33;1.68)** | **2.05 (1.11;2.92)** | **0.004** | 0.70(0.14;1.26) | 1.53(0.79;2.21) | 0.015 |
| **Vs CTIN +CINP** | | *-0.65(-1.30;0.01)* | -1.28(-2.08;-0.43) | 0.054 | | | | *-0.30(-1.03;0.43)* | -0.62 (-1.43;0.21) | 0.413 |
| **MAS 6PPT** | 4.22 (0.623) | 3.31(0.652) | | | 2.61(0.720) | | | 3.05(0.601) | | |
| **Vs HC** | | *0.91(0.18;1.64)* | 1.43 (0.77;2.06) | 0.015 | **1.61(0.70;2.51)** | **2.47 (1.46;3.39-)** | **0.001** | **1.17(0.42;1.92)** | **1.91 (1.12;2.62)** | **0.002** |
| **Vs CTIN +CINP** | | *-0.70(-1.58;0.18)* | -1.04(-1.83;-0.22) | 0.120 | | | | *-0.43(-1.41;0.54)* | -0.68 (-1.49;0.15) | 0.381 |
| **TRIG PPT** | 5.72 (0.808) | 5.27(0.846) | | | 3.61(0.934) | | | 4.68(0.779) | | |
| **Vs HC** | | *0.46(0.-0.49;1.40)* | 0.54 (-0.05;1.13) | 0.343 | **2.11(0.94;3.28)** | **2.50 (1.49;3.42)** | **0.001** | 1.04(0.07;2.01) | 1.31(0.60;1.97) | 0.036 |
| **Vs CTIN +CINP** | | ***-1.65(-2.79;-0.51)*** | **-1.90(-2.77;-0.98)** | **0.005** | | | | *-1.07(-2.33;0.20)* | -1.28(-2.13;-0.38) | 0.097 |
| **TRIG 6PPT** | 7.79 (0.973) | 6.22(1.018) | | | 4.51(1.124) | | | 6.05(0.938) | | |
| **Vs HC** | | ***1.57(0.44;2.71)*** | **1.58 (0.90;2.22)** | **0.007** | **3.28(1.87;4.69)** | **3.22 (2.08;4.25)** | **<0.001** | **1.75(0.58;2.91)** | **1.82 (1.04;2.52)** | **0.004** |
| **Vs CTIN +CINP** | | *-1.71(-3.08;-0.33)* | -1.63(-2.47;-0.74) | 0.015 | | | | *-1.54(-3.06;-0.01)* | -1.53(-2.40;-0.60) | 0.048 |
| **TA PPT** | 9.05 (1.273) | 9.98(1.332) | | | 6.43(1.470) | | | 8.63(1.227) | | |
| **Vs HC** | | *-0.94(-2.42;0.55)* | -0.72(-1.31;-0.11) | 0.215 | **2.61(0.77;4.46)** | **1.97 (1.05;2.81)** | **0.006** | *0.42(-1.11;1.95)* | 0.34 (-0.030;0.96) | 0.118 |
| **Vs CTIN +CINP** | | ***-3.55(-5.35;-1.75)*** | **-2.59(-3.52;-1.57)** | **<0.001** | | | | *-2.19(0.20;4.18)* | -1.67(-2.54;-0.74) | 0.031 |
| **TA 6PPT** | 12.61 (1.589) | 11.73(1.663) | | | 7.56(1.835) | | | 10.45(1.532) | | |
| **Vs HC** | | *0.89(-0.97;2.74)* | 0.54 (-0.05;1.13) | 0.347 | **5.05(2.75;7.36)** | **3.03 (1.94;4.02)** | **<0.001** | 2.16(0.25;4.07) | 1.38(0.66;2.05) | 0.027 |
| **Vs CTIN +CINP** | | ***-4.17(-6.41;-1.93)*** | **-2.43(-3.35;-1.45)** | **<0.001** | | | | *-2.89(-5.37;-0.41)* | -1.75(-2.64;-0.81) | 0.023 |
| **Lateral elbow PPT** | 7.16 (1.181) | 7.40(1.235) | | | 5.56(1.363) | | | 6.33(1.138) | | |

(*Continued*)

**Table 4.** (Continued）

| | HC (A) | CTIN (B) | | | CTIN + CINP (C) | | | CINP (D) | | |
|---|---|---|---|---|---|---|---|---|---|---|
| | Mean (SE) | Mean (SE), *MD (95% CI)** | SMD (95% CI)* | P | Mean (SE), *MD (95% CI)** | SMD (95% CI)* | P | Mean (SE), *MD (95% CI)** | SMD(95% CI)* | P |
| **Vs HC** | | *-0.24(-1.62;1.14)* | -0.20 (-0.78;0.38) | 0.727 | *1.60(-0.11;3.31)* | 1.29 (0.47;2.08) | 0.067 | *0.83(-0.59;2.25)* | 0.71(0.05;1.35) | 0.250 |
| **Vs CTIN +CINP** | | *-1.84(-3.51;-0.18)* | -1.44(-2.24;-0.59) | 0.031 | | | | *-0.77(-2.61;1.08)* | -0.63 (-1.42;0.19) | 0.411 |
| **Lateral elbow 6PPT** | 9.94 (1.591) | 9.19(1.664) | | | 6.71(1.837) | | | 8.13(1.533) | | |
| **Vs HC** | | *0.75(-1.11;2.61)* | 0.46 (-0.13;1.04) | 0.425 | **3.23(0.92;5.53)** | **1.94 (1.03;2.78)** | **0.007** | *1.82(-0.10;3.73)* | 1.16(0.46;1.81) | 0.062 |
| **Vs CTIN +CINP** | | *-2.47(-4.72;-0.23)* | -1.45(-2.25;-0.60) | 0.031 | | | | *-1.41(-3.90;1.08)* | -0.86(-1.66;-0.03) | 0.264 |
| **C5C6 HPT** | 45.37 (1.504) | 46.09(1.572) | | | 45.57(1.774) | | | 46.24(1.451) | | |
| **Vs HC** | | *-0.73(-2.48;1.03)* | -0.47 (-1.05;0.12) | 0.414 | *-0.20(-2.49;2.08)* | -0.13 (-0.93;0.68) | 0.860 | *-0.88(-2.68;0.93)* | -0.59 (-1.22;0.06) | 0.339 |
| **Vs CTIN +CINP** | | *0.52(-1.73;2.78)* | -0.32 (-1.12;0.49) | 0.647 | | | | *-0.67(-3.12;1.77)* | -0.43 (-1.27;0.43) | 0.587 |
| **C5C6 6HPT** | 48.79 (1.342) | 48.40(1.403) | | | 48.38(1.584) | | | 48.61(1.295) | | |
| **Vs HC** | | *0.39(-1.18;1.95)* | 0.29 (-0.30;0.86) | 0.627 | *0.41(-1.64;2.45)* | 0.29 (-0.52;1.09) | 0.407 | *0.17(-1.44;1.79)* | 0.14(-0.50;0.76) | 0.832 |
| **Vs CTIN +CINP** | | *-0.02(-2.03;1.99)* | -0.01 (-0.82;0.79) | 0.983 | | | | *-0.23(-2.41;1.95)* | -0.17 (-1.00;0.68) | 0.832 |
| **MAS HPT** | 45.74 (1.661) | 45.76(1.736) | | | 44.17(1.959) | | | 46.30(1.602) | | |
| **Vs HC** | | *-0.02(-1.96;1.92)* | 0.00 (-0.58;0.58) | 0.982 | *1.57(-0.96;4.09)* | 0.97 (0.12;1.79) | 0.222 | *-0.56(-2.55;1.44)* | -0.34 (-0.97;0.30) | 0.581 |
| **Vs CTIN +CINP** | | *-1.59(-4.08;0.90)* | -0.24 (-1.04;0.57) | 0.208 | | | | *-2.12(-4.82;0.57)* | -1.36(-2.25;-0.42) | 0.122 |
| **MAS 6HPT** | 48.78 (1.253) | 47.74(1.310) | | | 47.17(1.478) | | | 48.60(1.209) | | |
| **Vs HC** | | *1.04(-0.43;2.50)* | 0.81 (0.20;1.41) | 0.163 | *1.61(-0.30;3.52)* | 1.23 (0.36;2.06) | 0.097 | *0.18(-1.33;1.68)* | 0.15 (-0.49;0.77) | 0.818 |
| **Vs CTIN +CINP** | | *-0.57(-2.45;1.31)* | -0.42 (-1.22;0.39) | 0.547 | | | | *-1.43(-3.47;0.60)* | -1.10(-1.97;-0.19) | 0.166 |
| **TRIG HPT** | 44.59 (1.580) | 45.84(1.651) | | | 45.06(1.864) | | | 45.78(1.524) | | |
| **Vs HC** | | *-1.26(-3.10;0.59)* | -0.77(-1.37;-0.17) | 0.180 | *-0.47(-2.88;1.93)* | -0.28 (-1.08;0.52) | 0.697 | *-1.19(-3.09;0.71)* | -0.76(-1.40;-0.10) | 0.804 |
| **Vs CTIN +CINP** | | *-0.47(-2.88;1.93)* | -0.46 (-1.26;0.36) | 0.513 | | | | *-1.11(-3.13;0.92)* | -0.44 (-1.28;0.42) | 0.280 |
| **TRIG 6HPT** | 47.45 (1.246) | 47.46(1.303) | | | 47.23(1.470) | | | 48.34(1.202) | | |
| **Vs HC** | | *-0.02(-1.47;1.44)* | -0.01 (-0.59;0.57) | 0.984 | *0.22(-1.68;2.12)* | 0.17 (-0.64;0.97) | 0.819 | *-0.89(-2.39;0.61)* | -0.72(-1.36;-0.06) | 0.242 |
| **Vs CTIN +CINP** | | *-0.24(-2.10;1.63)* | -0.17 (-0.97;0.64) | 0.804 | | | | *-1.11(-3.13;0.92)* | -0.86 (-1.71;0.03) | 0.280 |
| **TA HPT** | 47.09 (1.260) | 48.11(1.317) | | | 45.96(1.486) | | | 47.67(1.22) | | |
| **Vs HC** | | *-1.02(-2.49;0.45)* | -0.79(-1.38,-0.19) | 0.171 | *1.13(-0.79;3.05)* | 0.85 (0.03;1.65) | 0.245 | *-0.58(-2.10;0.93)* | -0.47 (-1.09;0.18) | 0.446 |

(*Continued*)

**Table 4.** (Continued)

| | HC (A) | CTIN (B) | | | CTIN + CINP (C) | | | CINP (D) | | |
|---|---|---|---|---|---|---|---|---|---|---|
| | Mean (SE) | Mean (SE), *MD (95% CI)** | SMD (95% CI)* | P | Mean (SE), *MD (95% CI)** | SMD (95% CI)* | P | Mean (SE), *MD (95% CI)** | SMD(95% CI)* | P |
| **Vs CTIN +CINP** | | *-2.15(-4.04;-0.27)* | -1.57(-2.42;-0.68) | 0.026 | | | | *-1.71(-3.76;0.33)* | -1.30(-2.17;-0.39) | 0.100 |
| **TA 6HPT** | 49.29 (0.859) | 49.39(0.897) | | | 48.09(1.013) | | | 49.11(0.828) | | |
| **Vs HC** | | *-0.10(-1.10;0.90)* | -0.11 (-0.69;0.46) | 0.844 | *1.19(-0.11;2.50)* | 1.33 (0.46;2.15) | 0.073 | *0.173(-0.858;1.204)* | 0.21 (-0.42;0.84) | 0.740 |
| **Vs CTIN +CINP** | | *-1.29(-2.58;-0.01)* | -1.40(-2.22;-0.53) | 0.049 | | | | *-1.02(-2.41;0.38)* | -1.14(-2.00;-0.24) | 0.150 |
| **Lateral elbow HPT** | 45.93 (1.456) | 47.06(1.522) | | | 45.23(1.717) | | | 47.24(1.404) | | |
| **Vs HC** | | *-1.13(-2.83;0.57)* | -0.34 (-0.92;0.24) | 0.189 | *0.69(-1.52;2.91)* | 0.18 (-0.61;0.96) | 0.536 | *-1.31(-3.06;0.44)* | -0.37 (-1.00;0.27) | 0.141 |
| **Vs CTIN +CINP** | | *-1.83(-4.01;0.35)* | -1.16(-1.97;-0.32) | 0.099 | | | | *-2.00(-0.36;4.37)* | -1.33(-2.20;-0.41) | 0.096 |
| **Lateral elbow 6HPT** | 48.46 (0.995) | 48.16(1.040) | | | 47.40(1.174) | | | 48.20(0.960) | | |
| **Vs HC** | | *0.30(-0.86;1.46)* | 0.29 (-0.29;0.87) | 0.611 | *1.06(-0.46;2.57)* | 1.01 (0.18;1.81) | 0.170 | *0.26(-0.94;1.45)* | 0.27 (-0.37;0.89) | 0.669 |
| **Vs CTIN +CINP** | | *-0.76(-2.25;0.74)* | -0.71 (-1.49;0.10) | 0.317 | | | | *-0.80(-2.41;0.82)* | -0.77 (-1.60;0.09) | 0.331 |
| *Dynamic measures: conditioned pain modulation* | | | | | | | | | | |
| **Absolute CPM PPT** | 0.30 (0.455) | 0.45(0.477) | | | -0.03(0.522) | | | 0.04(0.439) | | |
| **Vs HC** | | *-0.15(-0.68;0.39)* | -0.32 (-0.90;0.27) | 0.588 | *0.33(-0.32;0.97)* | 0.69 (-0.09;1.44) | 0.320 | *0.26(-0.29;0.80)* | 0.58 (-0.07;1.21) | 0.353 |
| **Vs CTIN +CINP** | | *-0.47(-1.10;0.158)* | -0.97(-1.74;-0.18) | 0.140 | | | | *-0.07(-0.77;0.63)* | -0.15 (-0.93;0.64) | 0.846 |
| **Absolute CPM 6PPT** | 0.61 (0.732) | 0.43(0.766) | | | 0.30(0.840) | | | -0.09(0.706) | | |
| **Vs HC** | | *0.18(-0.68;1.03)* | 0.24 (-0.35;0.82) | 0.685 | *-0.31(-0.73;1.35)* | 0.41 (-0.35;1.15) | 0.557 | *0.70(-0.18;1.58)* | 0.97(0.29;1.62) | 0.119 |
| **Vs CTIN +CINP** | | *-0.13(-1.15;0.88)* | -0.16 (-0.91;0.59) | 0.794 | | | | *0.39(-0.74;1.51)* | 0.51 (-0.29;1.30) | 0.497 |
| **Relative CPM PPT** | 6.40 (5.412) | 11.67(6.582) | | | 5.01 (8.027) | | | 7.10(6.696) | | |
| **Vs HC** | | *-5.27(-17.77;7.23)* | -0.88(-1.47;-0.26) | 0.405 | *1.38(-13.83;16.60)* | 0.22 (-0.53;0.96) | 0.857 | *-0.70(-13.09;11.69)* | -0.12 (-0.74;0.51) | 0.911 |
| **Vs CTIN +CINP** | | *-6.65(-21.47;8.16)* | -0.95(-1.71;-0.15) | 0.375 | | | | *-2.08(-18.19;14.02)* | -0.29 (-1.08;0.51) | 0.798 |
| **Relative CPM 6PPT** | 4.52 (4.259) | 4.60(5.18) | | | 5.75(6.32) | | | 2.89(5.27) | | |
| **Vs HC** | | *-0.44(-10.28;9.40)* | -0.02 (-0.60;0.56) | 0.930 | *-1.23(-13.21;10.75)* | -0.25 (-0.99;0.50) | 0.839 | *1.63(-8.12;11.38)* | 0.35 (-0.29;0.97) | 0.741 |
| **Vs CTIN +CINP** | | *0.79(-10.86;12.45)* | 0.21 (-0.55;0.95) | 0.893 | | | | *2.86(-9.81;15.53)* | 0.50 (-0.30;1.29) | 0.656 |
| **Absolute CPM HPT** | 0.37 (0.788) | 0.46(0.823) | | | 0.47 (0.922) | -0.12 (-0.90;0.66) | | 0.76(0.760) | | |
| **Vs HC** | | *-0.10(-1.02;0.82)* | -0.11 (-0.69;0.47) | 0.834 | *-0.10(-1.27;1.07)* | | 0.864 | *-0.39(-1.34;0.56)* | -0.50 (-1.13;0.14) | 0.414 |
| **Vs CTIN +CINP** | | *0.01(-1.16;1.15)* | 0.01 (-0.77;0.79) | 0.994 | | | | *-0.29(-1.55;0.97)* | -0.35 (-1.17;0.48) | 0.649 |

(*Continued*)

**Table 4.** (Continued)

| | HC (A) | CTIN (B) | | | CTIN + CINP (C) | | | CINP (D) | | |
|---|---|---|---|---|---|---|---|---|---|---|
| | Mean (SE) | Mean (SE), *MD (95% CI)** | SMD (95% CI)* | P | Mean (SE), *MD (95% CI)** | SMD (95% CI)* | P | Mean (SE), *MD (95% CI)** | SMD(95% CI)* | P |
| **Absolute CPM 6HPT** | -0.06 (0.586) | 0.30(0.612) | | | -0.10(0.685) | | | 0.70(0.565) | | |
| **Vs HC** | | *-0.36(-1.04;0.33)* | -0.60 (-1.18;0.00) | 0.305 | *0.05(-0.83;0.92)* | 0.06 (-0.72;0.84) | 0.918 | *-0.76(-1.46;-0.05)* | -1.31(-1.98;-0.60) | 0.035 |
| **Vs CTIN +CINP** | | *-0.40(-1.26;0.45)* | -0.63 (-1.42;0.17) | 0.355 | | | | | *-0.80(-1.74;0.13)* | -1.32(-2.19;-0.40) | 0.091 |
| **Relative CPM HPT** | 2.47 (0.96) | 2.50(1.17) | | | 2.67(1.49) | | | 2.92(1.19) | | |
| **Vs HC** | | *-0.02(-2.24;2.19)* | -0.03 (-0.60;0.55) | 0.985 | *-0.19(-3.02;2.63)* | -0.18 (-0.96;0.61) | 0.892 | *-0.44(-2.64;1.76)* | -0.42 (-1.05;0.22) | 0.690 |
| **Vs CTIN +CINP** | | *0.17(-2.60;2.95)* | 0.13 (-0.65;0.91) | 0.902 | | | | *-0.25(-3.22;2.72)* | -0.19 (-1.01;0.63) | 0.868 |
| **Relative CPM 6HPT** | 1.04 (0.678) | 1.80(0.822) | | | 1.09(1.05) | | | 2.52(0.839) | | |
| **Vs HC** | | *-0.76(-2.32;0.80)* | -1.01(-1.61;-0.38) | 0.335 | *-0.05(-2.04;1.94)* | -0.06 (-0.84;0.72) | 0.960 | *-1.48(-3.03;0.07)* | -1.97(-2.70;-1.18) | 0.061 |
| **Vs CTIN +CINP** | | *-0.71(-2.67;1.24)* | -0.80 (-1.59;0.01) | 0.473 | | | | *-1.43(-3.52;0.66)* | -1.56(-2.46;-0.61) | 0.179 |

Data are presented as adjusted mean (standard error). Bold indicates significant results. All data were adjusted for age, sex, caffeine and nicotine consumption 24 hours prior to the QST measurements. Adjusted mean differences and 95% confidence interval are reported. P values were adjusted for multiple comparisons by means of a Bonferroni correction within models to reduce the risk of false-positive results. With adjustment, statistical significance was accepted at an alpha level of 0.010. Effect sizes (standardized mean differences) from 0.2 to 0.49 were considered small, 0.5 to 0.79 considered moderate, and 0.8 and above as large (124). Variables included in the same multivariate analysis of covariance model are marked in the same grey color. For several (6)HPT measurements, the 51° C threshold was reached before reaching the (6)HPT. This was the case for five to 17 measurements in the HC group, for three to 18 measurements in the CTIN group, for zero to seven measurements in the CTIN+CINP group and for four to 16 measurements in the CINP group, for the HPT and 6HPT respectively.

*Mean differences versus healthy controls were calculated as: A-B, A-C, A-D, mean differences versus patients with chronic tinnitus and chronic pain were calculated as: C-B and C-D.

HC: healthy controls, CTIN: patients with chronic tinnitus, CTIN+CINP: patients with chronic tinnitus and chronic idiopathic neck pain, CINP: patients with chronic idiopathic neck pain, SE: standard error, MD: mean difference, CI: confidence interval, SMD: standardized mean difference, C5-C6: articular pillar of the C5–C6 zygapophyseal joint, PPT: pressure pain threshold, 6PPT: 6/10 pressure pain threshold, HC: healthy controls, CTIN: chronic tinnitus, MAS: masseter muscle, TRIG: trigeminal nerve, TA: tibialis anterior muscle, HPT: heat pain threshold, 6HPT: 6/10 heat pain threshold, CPM: conditioned pain modulation

scores, compared to HCs but the difference was found to be the largest in the CTIN+CINP group, which also showed significantly higher scores compared to the CTIN and CINP groups. The cut-off for the presence of signs of CS (≥40/100) was reached in significantly more CTIN +CINP patients, compared to the other groups (Table 6).

**Self-reported psychological factors.** A univariate analysis of covariance showed a significant group difference for the CD-RISC25 (Table 5, Fig 3A). Patients with CTIN+CINP reported less resilience, compared to HCs and CTIN. A multivariate analysis of covariance showed significant group differences for the total DASS21, its subscales and BDI (Table 5, Fig 3A and 3B). Patients with CTIN+CINP reported higher depression, anxiety and stress levels, compared to HCs and CTIN. The same applies for patients with CINP, compared to HCs (P<0.001). Based on the cut-off score for the BDI (≥ 13/63), only the CTIN+CINP group consisted of significantly more patients with clinically significant depression, compared to HCs (Table 6).

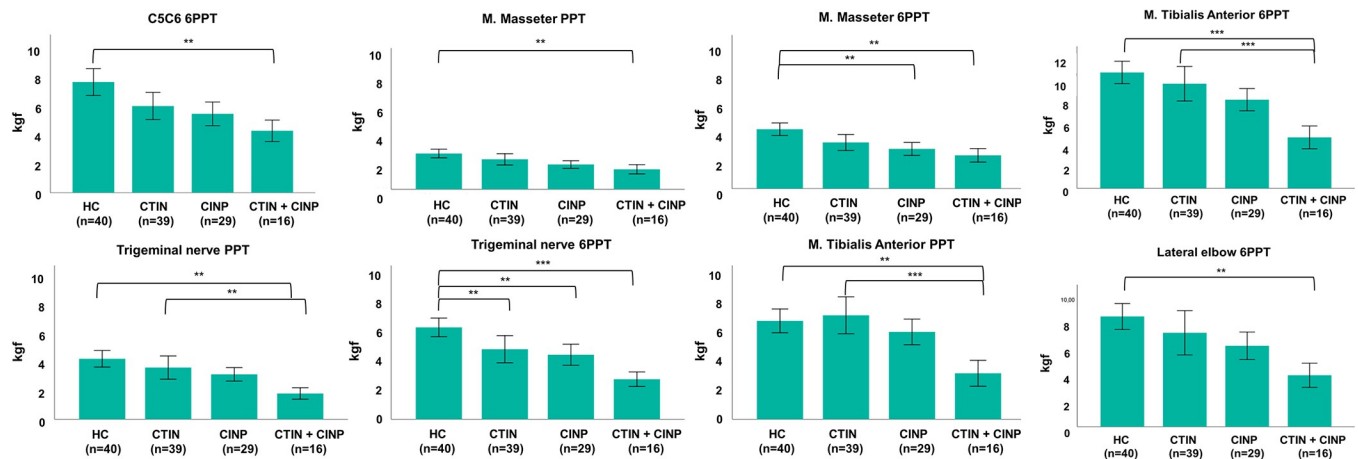

**Fig 2. A-B:** Bar plots of QST measures with significant group differences. C5-C6: articular pillar of the C5–C6 zygapophyseal joint, 6PPT: 6/10 pressure pain threshold, n: sample size, HC: healthy control, CTIN: chronic tinnitus, CINP: chronic idiopathic neck pain, PPT: pressure pain threshold, **: P<0.01, ***: P<0.001.

A multivariate analysis of covariance showed significant group differences for the BFI Extroversion and BFI Neuroticism (Table 5, Fig 3B). Post hoc pairwise comparisons revealed that patients with CTIN+CINP consider them being less extrovert, compared to HCs, whereas all patient groups reported to be more neurotic, compared to HCs. A univariate analysis of covariance displayed no significant group differences for PCS (Table 5).

**Self-reported lifestyle factors.** A univariate analysis of covariance showed no significant group differences for the Baecke Physical Activity questionnaire (Table 5). A multivariate analysis of covariance showed significant group differences for the PSQI and ISI (Table 5, Fig 3B). CTIN+CINP and CINP patients were shown to report significantly less perceived sleep quality and more severe insomnia complaints, compared to HCs. Sleep quality and insomnia severity were perceived significantly worse in CTIN+CINP patients, compared to CTIN but not compared to CINP. Based on the cut-off score for the PSQI ($\geq$ 5/21) and ISI ($\geq$ 15/28), significantly more patients with CTIN+CINP reported clinical insomnia, compared to HCs and CTIN. The same applies for patients with CINP, compared to HCs (Table 6).

**Correlations between signs and symptoms of CS and self-report tinnitus, pain, psychological and lifestyle measures.** Several significant, weak to strong correlations were found between signs and symptoms of CS (QST and CSI) and self-report questionnaires (tinnitus, pain, psychological and lifestyle measures) (Tables 7 and 8).

## Discussion

**The first aim of the present study was to compare signs and symptoms of CS between chronic tinnitus patients with and without CINP, patients with CINP only and HCs.** Patients with CTIN or CINP only, reported significantly more symptoms indicative for CS on the CSI, compared to HC. Both groups did show higher local mechanical hyperalgesia on at least one location in the trigeminal region, compared to HC. Interestingly, patients who suffered from CTIN+CINP reported significantly more symptoms indicative for CS on the CSI, compared to all other groups. In addition, they showed significantly higher local mechanical hyperalgesia at almost all locations, compared to HC and CTIN. To the best of our knowledge, this is the first study to evaluate CSI and QST measures in patients with tinnitus, prohibiting comparisons with other studies. The presence of mechanical hyperalgesia at the trigeminal

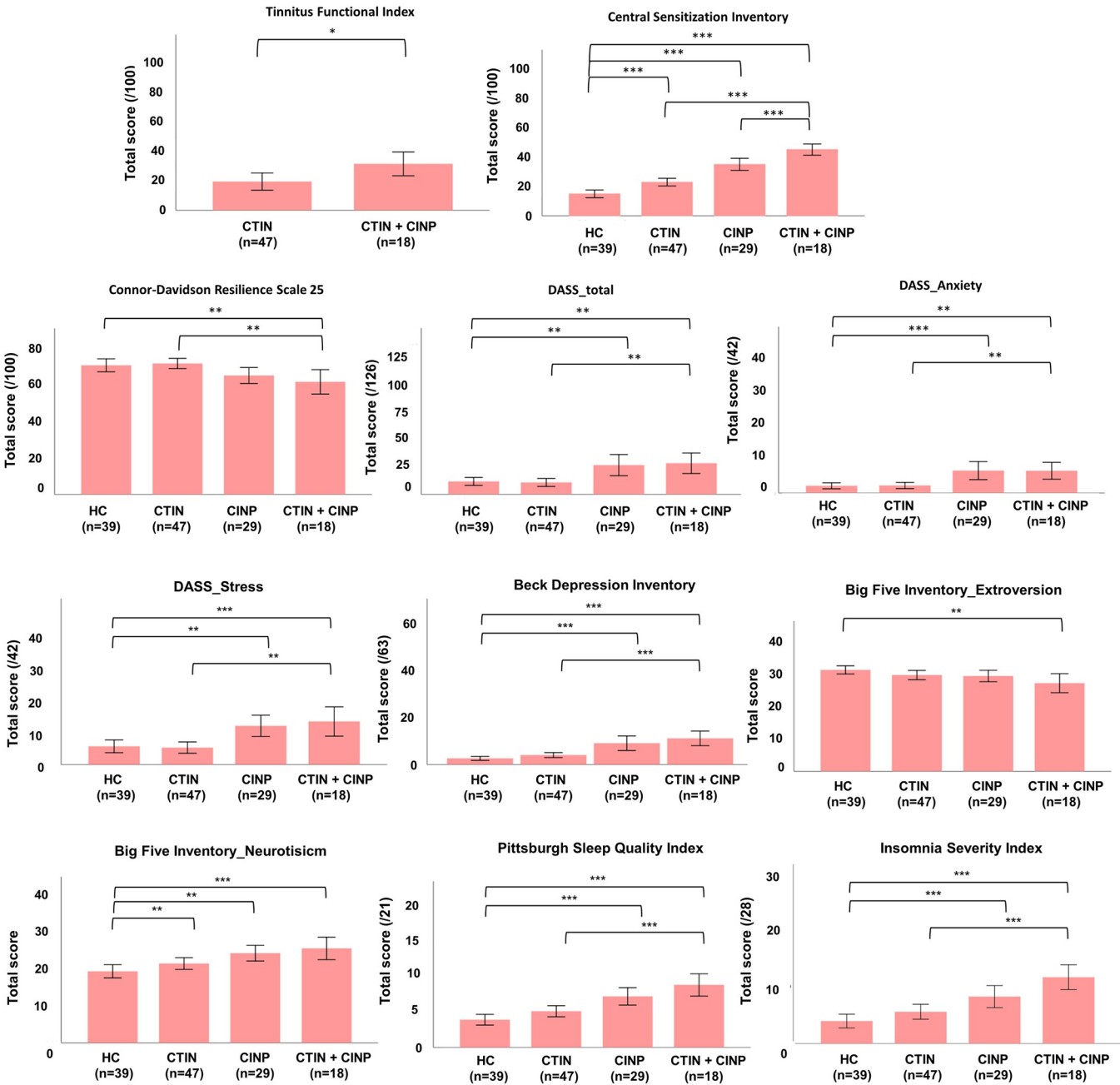

**Fig 3. A-B:** Bar plots of self-report measures with significant group differences. sample size, HC: healthy control, CTIN: chronic tinnitus, CINP: chronic idiopathic neck pain, DASS21: Depression, Anxiety and Stress scale, *: P<0.05, **: P<0.01, ***: P<0.001.

region in CTIN may be explained by neuronal interactions between trigeminal afferents and the central auditory system, for which also evidence has been found in animal studies [128–131]. The co-occurrence of CTIN and CINP seems to be associated with a larger area of local mechanical hyperalgesia than both conditions separately. The lower PPTs in the patient groups could also be explained by the presence of myofascial trigger points in craniocervical muscles, which have already been identified in neck pain patients with [132, 133] or without chronic tinnitus [134–136]. The significant mean differences of PPTs are comparable with the

**Table 5. Adjusted means and standard errors, and results of univariate and multivariate analysis of covariance ((M)ANCOVA) of self-report questionnaires.**

*Self-reported symptoms of central sensitization (baseline questionnaire)*

| | HC (A) | CTIN (B) | | | CTIN + CINP (C) | | | CINP (D) | | |
|---|---|---|---|---|---|---|---|---|---|---|
| | Mean (SE) | Mean (SE), MD (95% CI)* | SMD | P | Mean (SE), MD (95% CI)* | SMD | P | Mean (SE), MD (95% CI)* | SMD | P |
| **CSI (/100)** | 14.03 (1.396) | 23.79(1.312) | | | 43.32(2.05) | | | 32.93(1.658) | | |
| **Vs HC** | | **-9.76(-13.62;-5.90)** | **-8.61(-5.62;-7.44)** | **<0.001** | **-29.29(-34.16;-24.42)** | **-18.08 (-22.08;-13.46)** | **<0.001** | **-18.90(-23.12;-14.68)** | **-12.51 (-15.06;-9.53)** | **<0.001** |
| **Vs CTIN+CINP** | | **19.53 (14.63;24.43)** | **12.63 (9.54;15.34)** | **<0.001** | | | | **10.39 (5.26;15.53)** | **5.72 (3.94;7.30)** | **<0.001** |

*Self-reported psychological factors (baseline questionnaire)*

| | HC (A) | CTIN (B) | | | CTIN + CINP (C) | | | CINP (D) | | |
|---|---|---|---|---|---|---|---|---|---|---|
| | Mean (SE) | Mean (SE), MD (95% CI)* | SMD | P | Mean (SE), MD (95% CI)* | SMD | P | Mean (SE), MD (95% CI)* | SMD | P |
| **CD-RISC25 (/100)** | 69.14 (1.776) | 70.10(1.669) | | | 60.30(2.614) | | | 63.72(2.111) | | |
| **Vs HC** | | -0.96(-5.87;3.95) | -0.56 (-0.55;0.22) | 0.700 | **8.84 (2.64;15.05)** | **4.28 (2.96;5.45)** | **0.006** | 5.42 (0.06;10.79) | 2.82 (1.90;3.64) | 0.048 |
| **Vs CTIN+CINP** | | **-9.81(-16.04;-3.57)** | **-4.98(-6.22;-3.58)** | **0.002** | | | | -3.42 (-9.96;3.11) | -1.48(-2.32;-0.59) | 0.302 |
| **DASS21_total (/126)** | 10.38 (2.557) | 10.88(2.441) | | | 25.52(3.762) | | | 23.52(3.039) | | |
| **Vs HC** | | -0.50(-7.64;6.63) | -0.20 (-0.75;0.36) | 0.889 | **-15.13(-24.06;-6.21)** | **-5.09(-6.40;-3.60)** | **0.001** | **-13.14(-20.86;-5.42)** | **-4.75(-5.86;-3.64)** | **0.001** |
| **Vs CTIN+CINP** | | **14.63 (5.60;23.66)** | **5.12 (3.70;6.39)** | **0.002** | | | | 1.99 (-7.42;11.40) | 0.60 (-0.20;1.38) | 0.676 |
| **DASS21_Depression (/42)** | 3.67 (1.113) | 3.29(1.063) | | | 7.78(1.638) | | | 7.28(1.323) | | |
| **Vs HC** | | 0.38(-2.73;3.49) | 0.36 (-0.21;0.91) | 0.809 | -4.11(-8.00;-0.23) | -3.17(-4.16;-2.07) | 0.038 | -3.61(-6.97;-0.25) | -2.99(-3.84;-2.04) | 0.036 |
| **Vs CTIN+CINP** | | 4.49(0.56;8.42) | 3.61 (2.48;4.63) | 0.026 | | | | 0.50(-3.59;4.60) | 0.34 (-0.44;1.12) | 0.809 |
| **DASS21_Anxiety (/42)** | 1.84 (0.695) | 2.46(0.664) | | | 5.99(1.023) | | | 5.92(0.826) | | |
| **Vs HC** | | -0.62(-2.56;1.32) | -0.91(-1.48;-0.32) | 0.530 | **-4.15(-6.58,-1.73)** | **-5.13(-6.45;-3.63)** | **0.001** | **-4.08(-6.18;-1.98)** | **-5.42(-6.66;-4.00)** | **<0.001** |
| **Vs CTIN+CINP** | | **3.53(1.08;5.99)** | **4.54 (3.24;5.71)** | **0.005** | | | | 0.073 (-2.49;2.63) | 0.08 (-0.70;0.85) | 0.955 |
| **DASS21_Stress (/42)** | 4.87 (1.063) | 5.14(1.015) | | | 11.74(1.563) | | | 10.33(1.263) | | |
| **Vs HC** | | -0.26(-3.23;2.70) | -0.26 (-0.81;0.30) | 0.861 | **-6.87(-10.58;-3.16)** | **-5.56(-6.96;-3.96)** | **<0.001** | **-5.45(-8.66;-2.24)** | **-4.74(-5.86;-3.46)** | **0.001** |
| **Vs CTIN+CINP** | | **6.61 (2.85;10.36)** | **5.56 (4.04;6.90)** | **0.001** | | | | 1.42(-2.49;5.33) | 1.02 (0.18;1.82) | 0.474 |
| **BDI (/63)** | 2.43 (0.832) | 4.34(0.795) | | | 10.15(1.224) | | | 8.63(0.989) | | |
| **Vs HC** | | -1.92(-4.24;0.41) | -2.35(-3.03;-1.60) | 0.105 | **-7.73(-10.63;-4.82)** | **-7.98(-9.85;-5.83)** | **P<0.001** | **-6.21(-8.72;-3.69)** | **-6.88(-8.37;-5.15)** | **P<0.001** |
| **Vs CTIN+CINP** | | **5.81(2.87;8.75)** | **6.24 (4.59;7.71)** | **P<0.001** | | | | 1.52(-1.54;4.58) | 1.40 (0.52;2.23) | 0.328 |
| **BFI_Extroversion** | 29.92 (0.749) | 28.58(0.703) | | | 25.97(1.102) | | | 28.04(0.890) | | |
| **Vs HC** | | 1.34(-0.74;3.41) | 1.85 (1.16;2.48) | 0.204 | **9.95(1.33;6.56)** | **4.53 (3.16;5.75)** | **0.003** | 1.87(-0.39;4.14) | 2.32 (1.48;3.08) | 0.103 |
| **Vs CTIN+CINP** | | -2.61(-5.24;0.02) | -3.15(-4.10;-2.10) | 0.052 | | | | -2.07 (-4.83;0.68) | -2.12(-3.04;-1.13) | 0.139 |

*(Continued)*

**Table 5.** (Continued)

| | HC (A) | CTIN (B) | | | CTIN + CINP (C) | | | CINP (D) | | |
|---|---|---|---|---|---|---|---|---|---|---|
| | Mean (SE) | Mean (SE), MD (95% CI)* | SMD | P | Mean (SE), MD (95% CI)* | SMD | P | Mean (SE), MD (95% CI)* | SMD | P |
| **BFI _Agreeableness** | 34.07 (0.746) | 32.78(0.701) | | | 32.86(1.098) | | | 33.11(0.887) | | |
| **Vs HC** | | 1.29(-0.77;3.35) | 1.79 (1.11;2.41) | 0.218 | 1.21 (-1.39;3.82) | 1.39 (0.57;2.17) | 0.360 | 0.96(-1.29;3.22) | 1.19 (0.49;1.85) | 0.400 |
| **Vs CTIN+CINP** | | 0.080 (-2.54;2.70) | 0.10 (-0.62;0.81) | 0.952 | | | | -0.25 (-3.00;2.50) | -0.26 (-1.03;0.52) | 0.858 |
| **BFI _Conscientiousness** | 32.65 (0.805) | 32.66(0.756) | | | 31.94(1.184) | | | 31.94(0.956) | | |
| **Vs HC** | | -0.01(-2.24;2.22) | -0.01 (-0.57;0.54) | 0.993 | 0.71 (-2.10;3.52) | 0.76 (-0.01;1.50) | 0.617 | 0.71(-1.72;3.14) | 0.81 (0.15;1.45) | 0.563 |
| **Vs CTIN+CINP** | | -0.72(-3.55;2.11) | -0.81(-1.53;-0.06) | 0.615 | | | | $3.86*10^{-5}$(-2.96,2.96) | 0(-0.77;0.77) | 1.000 |
| **BFI _Neuroticism** | 18.73 (0.868) | 22.05(0.816) | | | 24.67(1.28) | | | 23.06(1.032) | | |
| **Vs HC** | | **-3.32(-5.72;-0.92)** | **-3.95(-4.83;-2.95)** | **0.007** | **-5.94(-8.97;-2.91)** | **-5.88(-7.34;-4.21)** | **P<0.001** | **-4.32(-6.95;-1.70)** | **-4.61(-5.70;-3.35)** | **0.001** |
| **Vs CTIN+CINP** | | 2.62(-0.43;5.67) | 2.72 (1.74;3.61) | 0.091 | | | | 1.61(-1.58;4.81) | 1.42 (0.54;2.26) | 0.320 |
| **BFI _Openness** | 36.03 (0.854) | 35.88(0.803) | | | 33.29(1.257) | | | 34.19(1.015) | | |
| **Vs HC** | | 0.16(-2.21;2.52) | 0.18 (-0.38;0.73) | 0.320 | 2.74 (-0.24;5.73) | 2.76 (1.73;3.69) | 0.071 | 1.84(-0.74;4.42) | 1.99 (1.19;2.72) | 0.160 |
| **Vs CTIN+CINP** | | -2.59(-5.59;0.41) | -2.73(-3.63;-1.76) | 0.090 | | | | -0.90 (-4.05;2.24) | -0.81 (-1.60;0.01) | 0.572 |
| **PCS (/52)** | NA | NA | NA | NA | 13.21(2.232) | NA | NA | 16.57(1.883) | | |
| **Vs CTIN+CINP** | | | | | | | | -3.57 (-8.96;2.25) | -1.66(-2.52;-0.75) | 0.234 |

*Self-reported lifestyle factors (baseline questionnaire)*

| | HC (A) | CTIN (B) | | | CTIN + CINP (C) | | | CINP (D) | | |
|---|---|---|---|---|---|---|---|---|---|---|
| **PSQI (/21)** | 3.84 (0.457) | 5.20(0.429) | | | 8.69(0.672) | | | 7.04(0.543) | | |
| **Vs HC** | | -1.36(-2.62;-0.10) | -3.08(-3.84;-2.22) | 0.035 | **-4.85(-6.45;-3.26)** | **-9.12 (-11.23;-6.70)** | **<0.001** | **-3.20(-4.58;-1.82)** | **-6.47(-7.89;-4.83)** | **<0.001** |
| **Vs CTIN+CINP** | | **3.49(1.89;5.10)** | **6.89 (5.10;8.48)** | **<0.001** | | | | 1.65(-0.03;3.33) | 2.77 (1.67;3.78) | 0.054 |
| **ISI (/28)** | 3.96 (0.717) | 5.39(0.674) | | | 11.59(1.055) | | | 8.28(0.852) | | |
| **Vs HC** | | -1.43(-3.41;0.55) | -2.10(-2.76;-1.39) | 0.156 | **-7.63(-10.13;-5.13)** | **-9.15 (-11.26;-6.72)** | **<0.001** | **-4.32(-6.48;-2.15)** | **-5.57(-6.82;-4.12)** | **<0.001** |
| **Vs CTIN+CINP** | | **6.20(3.68;8.72)** | **7.96 (5.93;9.75)** | **<0.001** | | | | 3.31(0.67;5.95) | 3.54 (2.28;4.69) | 0.014 |
| **Baecke Q (/15)** | 8.17 (0.217) | 7.64(0.204) | | | 7.91(0.320) | | | 7.56(0.258) | | |
| **Vs HC** | | 0.53(-0.07;1.13) | 2.52 (1.75;3.22) | 0.085 | 0.26 (-0.50;1.01) | 1.03 (0.24;1.78) | 0.505 | 0.61(-0.05;1.26) | 2.60 (1.71;339) | 0.069 |

(*Continued*)

**Table 5.** (Continued)

|  | HC (A) | CTIN (B) | | | CTIN + CINP (C) | | | CINP (D) | | |
|---|---|---|---|---|---|---|---|---|---|---|
|  | Mean (SE) | Mean (SE), MD (95% CI)* | SMD | P | Mean (SE), MD (95% CI)* | SMD | P | Mean (SE), MD (95% CI)* | SMD | P |
| Vs CTIN+CINP |  | 0.27(-0.49;1.03) | 1.12 (0.35;1.86) | 0.484 |  |  |  | 0.35(-0.45;1.15) | 1.24 (0.38;2.05) | 0.387 |

Data are presented as adjusted mean (standard error). Bold indicates significant results. All data were adjusted for sex. Adjusted mean differences and 95% confidence interval are reported. P values were adjusted for multiple comparisons by means of a Bonferroni correction within models to reduce the risk of false-positive results. With adjustment, statistical significance was accepted at an alpha level of 0.010. Effect sizes from 0.2 to 0.49 were considered small, 0.5 to 0.79 considered moderate, and 0.8 and above as large (124). Variables included in the same multivariate analysis of covariance model are marked in grey color.

*(Standardized) mean differences versus healthy controls were calculated as: A-B, A-C, A-D, mean differences versus patients with chronic tinnitus and chronic pain were calculated as: C-B and C-D.

HC: healthy controls, CTIN: patients with chronic tinnitus, CTIN+CINP: patients with chronic tinnitus and chronic idiopathic neck pain, CINP: patients with chronic idiopathic neck pain, SE: standard error, MD: mean difference, CI: confidence interval, SMD: standardized mean difference, TFI: Tinnitus Functional Index, HQ: Hyperacusis Questionnaire, NDI: Neck Disability Index, PCS: Pain Catastrophizing Scale, CSI: Central Sensitization Inventory, CD-RISC25:, Connor-Davidson Resilience Scale 25, DASS21: Depression, Anxiety and Stress Scale, BDI: Beck Depression Inventory, BFI: Big Five Index, PSQI: Pittsburgh Sleep Quality Index, ISI: Insomnia Severity Index, Q: questionnaire

suggested required differences for real change (at least 1.77 kgf/cm$^2$ [137]) or minimal clinical important difference (between 0.51 kgf/cm$^2$ and 2,24 kgf/cm$^2$ [138]) for PPT.

No differences in distant hyperalgesia were found between the CTIN, CINP group, and HC group, whereas the co-occurrence of CTIN and CINP was associated with higher distant mechanical hyperalgesia at all locations, compared to HC. Distant hyperalgesia is commonly regarded as an indicator of CS [61, 62] and involves an increased sensitivity located beyond the site of injury, which could be explained by an expansion of receptive fields at the spinal cord [139]. It should be noted that several patients in the CTIN+CINP suffered from chronic pain at one or more additional locations, which was not the case in the CINP group. A higher degree of CS may be expected in patients with widespread pain [61]. **The second aim was to compare psychological and lifestyle factors between all groups.** Apart from a higher degree of neuroticism, patients with CTIN did not show differences in psychological and lifestyle factors, compared to HC. Patients with CTIN+CINP reported lower resilience, higher levels of anxiety, stress and depression, and rated themselves as less extroverted and more neurotic, compared to HC and CTIN. Our results are in line with several studies in which significantly more anxiety [140, 141], stress [140–142] and depression [9, 140–144] has been found in patients with tinnitus and pain, compared to patients with tinnitus only. In contrast, other studies show less differences between tinnitus patients with and without pain regarding anxiety, stress and depression in subgroups with the same level of tinnitus impact [140, 141]. Neuroticism has been shown to be associated with the prevalence and severity of tinnitus [145] but studies comparing tinnitus with tinnitus and comorbid pain are lacking. The presence of higher psychological distress in CTIN+CINP and CINP can be explained by involvement of the limbic brain system or the autonomic nervous system for which evidence has been found in patients with tinnitus [146] and patients with chronic pain [147]. In our study, patients with CTIN+CINP also reported lower sleep quality and a higher degree of insomnia, compared to HC and CTIN. Other studies also show associations between tinnitus and subjective (PSQI) [36, 148] and objective (polysomnography) [148] sleep disturbances but studies comparing tinnitus with tinnitus and comorbid pain is lacking. Patients with CTIN+CINP generally showed similar deficits in psychological measures and sleep as CINP, but showed a higher psychological burden and poorer sleep, compared to CTIN. Interestingly, significantly more

**Table 6. Between-group comparisons for categorical variables of self-report questionnaires.**

*QST measures*

| Outcome measure | Group | Frequencies (n (%)) | | | P-value | P-value post hoc |
|---|---|---|---|---|---|---|
| **CPM PPT**[b] | | *Responder* | *Non responder* | | 0.539 | NA |
| | *HC* | 31 (77.5) | 9(22.5) | | | |
| | *CTIN* | 33(84.6) | 6(15.4) | | | |
| | *CTIN +CINP* | 15(88.2) | 2(11.8) | | | |
| | *CINP* | 21(72.4) | 8(27.6) | | | |
| **CPM 6PPT**[b] | | *Responder* | *Non responder* | | 0.519 | NA |
| | *HC* | 28(70.0) | 12(30.0) | | | |
| | *CTIN* | 30(76.9) | 9(23.1) | | | |
| | *CTIN +CINP* | 15(88.2) | 2(11.8) | | | |
| | *CINP* | 21(72.4) | 8(27.6) | | | |
| **CPM HPT**[b] | | *Responder* | *Non responder* | | 0.977 | NA |
| | *HC* | 32(80.0) | 8(20.0) | | | |
| | *CTIN* | 32(80.0) | 8(20.0) | | | |
| | *CTIN +CINP* | 12(80.0) | 3(20.0) | | | |
| | *CINP* | 22(75.9) | 7(24.1) | | | |
| **CPM 6HPT**[b] | | *Responder* | *Non responder* | | 0.514 | NA |
| | *HC* | 27(67.5) | 13(32.5) | | | |
| | *CTIN* | 31(77.5) | 9(22.5) | | | |
| | *CTIN +CINP* | 11(73.3) | 4(26.7) | | | |
| | *CINP* | 24(82.8) | 5(17.2) | | | |

*Self-report tinnitus measures*

| Outcome measure | Group | Frequencies (n (%)) | | | P-value | P-value post hoc |
|---|---|---|---|---|---|---|
| **TFI**[b] | | *Mild (0–25)* | *Significant (26–50)* | *Severe (>51)* | **0.027** | NA |
| | *CTIN* | 35(74.5) | 8(17.0) | 4(8.5) | | |
| | *CTIN +CINP* | 8(44.4) | 9(50.0) | 1(5.6) | | |
| **HQ**[b] | | *No hyperacusis (<28.4)* | *Hyperacusis (>28.4)* | | 0.675 | NA |
| | *CTIN* | 42(89.4) | 5(10.6) | | | |
| | *CTIN +CINP* | 15(83.3) | 3(16.7) | | | |

*Self-report symptoms of central sensitization*

| Outcome measure | Group | Frequencies (n (%)) | | | P-value | P-value post hoc |
|---|---|---|---|---|---|---|
| **CSI**[b] | | *No central sensitization (<40)* | *Central sensitization (>40)* | | **<0.001** | HC vs CTIN (0.123) |
| | *HC* | 39(100) | 0(0) | | | **HC vs CTIN + CINP (<0.001)** |
| | *CTIN* | 43(91.5) | 4(8.5) | | | **HC vs CINP (0.002)** |
| | *CTIN +CINP* | 6(33.3) | 12(66.7) | | | **CTIN vs CTIN + CINP (<0.001)** |
| | *CINP* | 22(75.9) | 7(24.1) | | | **CINP vs CTIN + CINP (0.004)** |
| **BDI**[b] | | *No depression (<13/63)* | *Depression (≥13/63)* | | **0.001** | HC vs CTIN (0.246) |
| | *HC* | 39(100) | 0(0) | | | **HC vs CTIN + CINP (0.001)** |
| | *CTIN* | 43(93.5) | 3(6.5) | | | HC vs CINP (0.073) |
| | *CTIN +CINP* | 13(68.4) | 6(31.6) | | | CTIN vs CTIN + CINP (0.015) |
| | *CINP* | 26(89.7) | 3(10.3) | | | CINP vs CTIN + CINP (0.127) |

*(Continued)*

**Table 6.** (Continued)

| Outcome measure | Group | Frequencies (n (%)) | | | P-value | P-value post hoc |
|---|---|---|---|---|---|---|
| PSQI[a] | | *No clinically significant sleep problems (<5)* | *Clinically significant sleep problems (≥5)* | | <0.001 | HC vs CTIN (0.082) |
| | HC | 26(66.7) | 13(33.3) | | | **HC vs CTIN + CINP (<0.001)** |
| | CTIN | 22(46.8) | 25(53.2) | | | **HC vs CINP (0.001)** |
| | CTIN +CINP | 2(11.1) | 16(88.9) | | | **CTIN vs CTIN + CINP (0.009)** |
| | CINP | 7(24.1) | 22(75.9) | | | CINP vs CTIN + CINP (0.449) |
| ISI[b] | | No clinical insomnia(0–7) | Subthreshold insomnia (8–14) | Clinical insomnia(15–21) | <0.001 | HC vs CTIN (0.351) |
| | HC | 33(84.6) | 6(15.4) | 0(0) | | **HC vs CTIN + CINP (<0.001)** |
| | CTIN | 34(72.3) | 12(25.5) | 1(2.1) | | **HC vs CINP (0.006)** |
| | CTIN +CINP | 4(22.2) | 9(50.0) | 5(27.8) | | **TIN vs CTIN + CINP (<0.001)** |
| | CINP | 16(55.2) | 9(31.0) | 4(13.8) | | CINP vs CTIN + CINP (0.076) |

[a]Chi-Square test

[b]Fisher's Exact Test

QST: Quantitative Sensory Testing, n: sample size, CPM: conditioned pain modulation, PPT: pressure pain threshold, 6PPT: 6/10 pressure pain threshold, HC: healthy controls, CTIN: patients with chronic tinnitus, CTIN+CINP: patients with chronic tinnitus and chronic idiopathic neck pain, CINP: patients with chronic idiopathic neck pain, NA: not applicable, C5-C6: articular pillar of the C5–C6 zygapophyseal joint, HPT: Heat Pain Threshold, 6HPT: 6/10 Heat Pain Threshold, TFI: Tinnitus Functional Index, HQ: Hyperacusis Questionnaire, CSI: Central Sensitization Inventory, BDI: Beck Depression Inventory, PSQI: Pittsburgh Sleep Quality Index, ISI: Insomnia Severity Index

patients with CTIN+CINP suffered from significant tinnitus (TFI > 26), compared to the CTIN group and the mean difference of 13.20 points is in line with the proposed MCID ranging from 13 [89] to 14 points [149]. It can be hypothesized that psychological and sleep problems are associated with the level of tinnitus impact, which is in line with several studies showing a higher psychological burden [35, 150–152] and more sleeping problems [150–152] in patients with higher tinnitus impact. Another possibility is that psychological burden and poor sloop are rather associated with pain than with tinnitus. **A third aim was to explore the relationship between signs and symptoms of CS on the one hand, and self-reported tinnitus, pain, psychological and lifestyle measures on the other hand**. In the present study, significant associations were found between signs and symptoms of CS, psychological factors and sleep in the whole group. This is in line with other studies in patients with traumatic chronic neck pain [153], peripheral joint pain [51] and healthy people [154, 155]. Also, tinnitus impact showed significant associations with symptoms (CSI) but not with signs (QST) of CS. Similarly, several studies show that tinnitus impact is associated with pain experience [9, 140, 141, 156], which may give rise to the hypothesis that chronic pain can be a risk factor for the development of more severe tinnitus or vice versa [140]. Also, stronger associations are found between psychological factors and pain perceptions in patients with higher tinnitus impact [35], and tinnitus impact has been shown to increase with an increasing number of pain sites [142]. No significant differences or associations were found for the presence or severity of hyperacusis. Debate exists about the validity of the HQ in patients with tinnitus [157] and about the cut-off point for clinical diagnosis of hyperacusis [158], which might influence

**Table 7. Correlations between signs and symptoms of central sensitization and self-report questionnaires in the total patient group.**

Total group (CTIN, CTIN + CINP and CINP group)

| | CSI | C5-C6 PPT | C5-C6 6PPT | MAS PPT | MAS 6PPT | MAS 6HPT | TRIG PPT | TRIG 6PPT | TA PPT | TA 6PPT | TA HPT | TA 6HPT | LE PPT | LE 6PPT | Abs CPM 6HPT | Rel CPM 6HPT |
|---|---|---|---|---|---|---|---|---|---|---|---|---|---|---|---|---|
| CD-RISC25 | r = -0.354 (P<0.001) | | | | | | | | | | | | | | | |
| DASS21_D | r = 0.481 (P<0.001) | | | | | | | | | | | | | | | |
| DASS21_A | r = 0.578 (P<0.001) | r = -0.233 (P = 0.010) | r = -0.257 (P = 0.004) | | r = -0.237 (P = 0.009) | r = -0.239 (P = 0.009) | r = -0.239 (P = 0.008) | | | | | | | | | |
| DASS21_S | r = 0.638 (P<0.001) | | r = -0.238 (P = 0.008) | | | | | | r = -0.253 (P = 0.005) | | r = -0.253 (P = 0.005) | | | | | |
| DASS21_Total | r = 0.625 (P<0.001) | | r = -0.247 (P = 0.006) | | | | | | | | r = -0.233 (P = 0.010) | r = -0.321 (P<0.001) | | | | |
| BDI | r = 0.644 (P<0.001) | | r = -0.262 (P = 0.004) | r = -0.254 (P = 0.005) | r = -0.289 (P = 0.001) | | | r = -0.262 (P = 0.004) | r = -0.264 (P = 0.003) | r = -0.325 (P<0.001) | | | | r = -0.270 (P = 0.003) | | |
| BFI_Extroversion | r = -0.249 (P = 0.005) | | | | | | | | | | | | | | | |
| BFI_Neuroticism | r = 0.496 (P<0.001) | | r = -0.261 (P = 0.004) | | r = -0.259 (P = 0.004) | r = -0.263 (P = 0.004) | | r = -0.282 (P = 0.002) | | r = -0.334 (P<0.001) | | r = -0.238 (P = 0.008) | | r = -0.271 (P = 0.002) | r = 0.244 (P = 0.007) | r = 0.236 (P = 0.009) |
| PSQI | r = 0.610 (P<0.001) | r = -0.265 (P = 0.003) | r = -0.352 (P<0.001) | r = -0.294 (P = 0.001) | r = -0.350 (P<0.001) | r = -0.252 (P = 0.005) | r = -0.306 (P = 0.001) | r = -0.381 (P<0.001) | r = -0.290 (P = 0.001) | r = -0.402 (P<0.001) | | | r = -0.236 (P = 0.009) | r = -0.348 (P<0.001) | | |
| ISI | r = 0.567 (P<0.001) | r = -0.265 (P = 0.003) | r = -0.328 (P<0.001) | r = -0.254 (P = 0.005) | r = -0.285 (P = 0.001) | | r = -0.296 (P = 0.001) | r = -0.357 (P<0.001) | r = -0.304 (P = 0.001) | r = -0.393 (P<0.001) | | | r = -0.267 (P = 0.003) | r = -0.336 (P<0.001) | | |

**Table 8. Correlations between signs and symptoms of central sensitization and self-report questionnaires in the CTIN + CINP and CINP group (pain-related measures), and CTIN and CTIN + CINP group (tinnitus-related measures).**

|  | CSI |
|---|---|
|  | CTIN+CINP & CINP group |
| **NDI** | r = 0.510 |
|  | (P = 0.031) |
|  | CTIN & CTIN + CINP group |
| **TFI** | r = -0.314 |
|  | (P = 0.016) |

CSI: Central Sensitization Inventory, C5-C6: articular pillar of the C5–C6 zygapophyseal joint, PPT: Pressure Pain Threshold, 6PPT: Pressure Pain Threshold 6/10, MAS: Masseter muscle, HPT: Heat Pain threshold, 6HPT: Heat Pain Threshold 6/10, TRIG: Trigeminal nerve, TA: Tibialis Anterior muscle, LE: lateral elbow, abs CPM: absolute CPM effect, rel CPM: relative CPM effect, CD-RISC25: Connor-Davidson Resilience Scale 25, DASS21: Depression, Anxiety and Stress Scale, BDI: Beck Depression Inventory, BFI: Big Five Inventory, PSQI: Pittsburgh Sleep Quality Index, ISI: Insomnia Severity Index, CTIN+CINP: patients with chronic tinnitus and chronic idiopathic neck pain, CINP: patients with chronic idiopathic neck pain, CTIN: patients with chronic tinnitus, NDI: Neck Disability Index, TFI: Tinnitus Functional Index

results. Associations between pain-related disability and CSI were found, which contrasts with the meta-analysis of Hübscher et al. (2013) [159].

The present study can be considered an innovative study since it was the first to assess possible signs and symptoms of CS in patients with chronic tinnitus. However, several limitations should be taken into account. First, the inclusion of patients was based on self-reported information. Since both chronic tinnitus and pain are complex and heterogenous conditions, clinical examination before inclusion could be warranted to allow for a clinically representative setting. Also, self-report may increase the risk of recall bias. In addition, only a very low proportion of the CTIN patients was recruited from ENT departments, which may explain the low percentage of patients with significant to severe TIN. This may have caused selection bias, not fully representing the general tinnitus population and causing an underestimation of tinnitus impact. Second, since this is a cross-sectional study, no conclusions about causality could be made. Third, the assessors were not blinded to the participant's condition, since participants in the HC or CTIN group had to be excluded when they reported pain of more than 2/10 on the day of testing. Fourth, several participants reached the maximal temperature of 51°C before reaching the (6)HPT and thus did not reach their actual heat pain thresholds. Fifth, since no gold standard for CS evaluation exists, no information about sensitivity and specificity of QST measurers is available. Future longitudinal studies should help us to better understand the complex interaction between chronic tinnitus, pain and CS. Sixth, when interpreting the results, the large number of measures should be considered since this may lead to chance findings from multiple comparisons. However, by performing MANCOVA analyses on clusters of correlated outcome measures, we aimed to reduce independent tests and account for interrelated measures. Also, future studies should investigate the added value of implementing treatment methods for chronic pain (such as pain neuroscience education and exercise therapy) in the treatment program of patients with tinnitus and comorbid pain.

## Conclusion

The present study identified preliminary evidence for the presence of CS in patients with chronic tinnitus when this is accompanied with CINP. These patients also experienced more

tinnitus impact, psychological burden and sleep problems than tinnitus patients without CINP. Signs and symptoms of CS were also shown to be associated with tinnitus impact, pain related disability, psychological burden and sleep disturbances. These results stress the importance of screening for comorbid CINP (and possibly other types of chronic pain) in patients with chronic tinnitus.

## Supporting information

**S1 Table. Standardization of test locations for pressure pain and heat pain thresholds.** (DOCX)

**S1 File. Ethical committee protocol English.** (PDF)

**S2 File. Ethical committee protocol Dutch.** (DOCX)

## Acknowledgments

We would like to express our sincere gratitude to the patients and healthy controls that participated in this study.

## Author Contributions

**Conceptualization:** Kayleigh De Meulemeester, Mira Meeus, Barbara Cagnie, Hannah Keppler.

**Data curation:** Kayleigh De Meulemeester, Robby De Pauw, Dorine Lenoir.

**Formal analysis:** Kayleigh De Meulemeester, Robby De Pauw.

**Investigation:** Kayleigh De Meulemeester, Dorine Lenoir.

**Methodology:** Kayleigh De Meulemeester, Mira Meeus, Robby De Pauw, Barbara Cagnie, Hannah Keppler, Dorine Lenoir.

**Project administration:** Kayleigh De Meulemeester, Dorine Lenoir.

**Supervision:** Mira Meeus, Barbara Cagnie, Hannah Keppler.

**Writing – original draft:** Kayleigh De Meulemeester.

**Writing – review & editing:** Mira Meeus, Robby De Pauw, Barbara Cagnie, Hannah Keppler, Dorine Lenoir.

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
