## [Decision Letter · Decision Letter 0]

25 Apr 2023

PONE-D-23-01457Suffering from chronic tinnitus, chronic neck pain, or both: does it impact the presence of signs and symptoms of central sensitization?PLOS ONE

Dear Dr. De Meulemeester,

Thank you for submitting your manuscript to PLOS ONE. After careful consideration, we feel that it has merit but does not fully meet PLOS ONE’s publication criteria as it currently stands. Therefore, we invite you to submit a revised version of the manuscript that addresses the points raised during the review process. Please address both reviewers' comments below.

We look forward to receiving your revised manuscript.

Kind regards,

Bianka Karshikoff, PhD

Academic Editor

PLOS ONE

Journal Requirements:

Reviewers' comments:

Reviewer's Responses to Questions

**Comments to the Author**

1. Is the manuscript technically sound, and do the data support the conclusions?

Reviewer #1: Yes

Reviewer #2: Yes

2. Has the statistical analysis been performed appropriately and rigorously? 

Reviewer #1: Yes

Reviewer #2: Yes

3. Have the authors made all data underlying the findings in their manuscript fully available?

Reviewer #1: Yes

Reviewer #2: Yes

4. Is the manuscript presented in an intelligible fashion and written in standard English?

Reviewer #1: Yes

Reviewer #2: Yes

5. Review Comments to the Author

Reviewer #1: Authors present results from a cross-sectional study that assesses central sensitization, tinnitus, pain, psychological factors and lifestyle factors in individuals with tinnitus (with and without chronic idiopathic neck pain), individuals with chronic idiopathic neck pain only, and healthy controls. Different measures are compared between groups and correlations are also assessed. The manuscript will be strengthened if the authors consider the following points.

1. Authors should include information about nicotine use and alcohol use in Table 3 (results are presented in lines 363-364 without any supporting data).

2. Table 3: Why change the definition of (standardized) mean differences across measures? That may cause some confusion.

3. While the authors do account for multiple comparisons within an outcome measure, they do not do anything to account for the large number of outcomes being assessed.

Minor points:

1. lines 149-150: authors might consider rewriting +/- 2 hours and +/- 1 hour and instead give the approximate time it takes to complete the assessments.

2. lines 305-306: "Total score ranges..." is stated in lines 302-303.

3. Authors should clarify the number of participants included in the study. line 350 indicates 48 patients with tinnitus only, while Figure 1 has 47. Numbers in Table 3 also differ from what is stated in line 350 and Figure 1. For example, for Sex, there is information presented for 19 CTIN+CINP, but line 350 says there are only 18. Also, the numbers presented for (dominant) painful side do not seem correct for CINP (and the percentage seems incorrect for Left for CTIN+CINP). Authors should carefully check the numbers in Table 3 and make sure they are consistent within the table as well as with what is presented in Figure 1 and in the text.

4. lines 369-370: Authors refer to Table 5 twice, but I believe this should be Table 3

5. lines 377: authors refer to differences in the Masseter muscle (PTT) and C5-C6-location (6PPT) between CINP and HC, but these are not significant according to their stated criteria.

6. Figures 2 and 3 are described as histograms, but histograms are not actually presented.

7. line 397: change "CTIN and CINP group" to "CTIN and CINP groups"

8. line 414: authors refer to Fig 3 for the Physical Activity, but this measure is not shown in the figure.

Reviewer #2: De Meulemeester and colleagues compared (1) signs and symptoms indicative for central sensitization using self-report Central Sensitization Inventory and QST and (2) psychological lifestyle factors using self-report questionnaires, and (3) the relationship between central sensitization, psychological and lifestyle variables and self-reported tinnitus and pain in tinnitus patients with and without chronic idiopathic neckpain, in patients with chronic idiopathic neckpain only and in healthy controls (HC). The manuscript is very well-written. In particular, the statistical methods section is very detailed: normality tests were used, followed by the appropriate parametric or non-parameteric statistical tests, correction for multiple comparisons were performed. In case of group-differences in demographical variables, these were appropriately controlled for. Age and sex were controlled for throughout the QST models.

Regarding the first aim, The authors found that symptoms of central sensitization were most pronounced in the group with comorbid chronic idiopathic neck pain and tinnitus. Regarding the second aim, patients with tinnitus and pain reported higher levels of anxiety, depression, stress and neuroticism and lower levels of resilience. Regarding the third aim, tinnitus impact was significantly associated with self-report measures of central sensitization but not QST.

There are two versions of the same manuscript in the .pdf file. Below, I comment on the manuscript named "Manuscript_Resubmission" .

1. Please consider rephrasing the third aim in the introduction (ln 112-114) to make it more comprehensible.

3. Instead of using “/” as a separator in the tables, consider making individual columns.

4. Page 32, first paragraph, ln 2 from bottom: make sure “and QST measures” should be removed. (the tables are displayed before the discussion, and the pages lost their numerical count on the left handside)

5. The figure quality and resolution is extremely poor across all figures. The images are blurry and it is impossible to read the numbers and letters on the x and y axis. I suggest the authors upload new figures in better resolution and also increase the font size on the x and y axis.

6. PLOS authors have the option to publish the peer review history of their article (what does this mean?). If published, this will include your full peer review and any attached files.

Reviewer #1: No

Reviewer #2: No

---

## [Author Response · Author response to Decision Letter 0]

12 Jun 2023

COVER LETTER WITH REPLY TO THE REMARKS OF THE REVIEWERS

We express our sincere gratitude to the reviewers for their valuable feedback and constructive comments, which have greatly contributed to the improvement of this manuscript. In this correspondence, we have included a comprehensive list of all newly added or modified sentences in the manuscript, clearly highlighted in yellow. We have also indicated the corresponding line numbers where these changes have been made in the manuscript. In the manuscript itself, newly added sentences or modified sentences are also highlighted in yellow. Furthermore, any sections that have been removed from the manuscript are identified through comments indicating the removal of the respective text. We hope that the modifications made in this revised version, as well as the responses provided to the questions raised, meet the expectations of the reviewers.

1. Responses to reviewer 1

1. Authors should include information about nicotine use and alcohol use in Table 3 (results are presented in lines 363-364 without any supporting data).

Thank you for this helpful suggestion. We added the data regarding the compliance with the instructions before QST measurements (nicotine use, alcohol use, caffeine use, vigorous physical activity and intake of pain medication) to Table 3 (last rows).

2. Table 3: Why change the definition of (standardized) mean differences across measures? That may cause some confusion.

Thank you for this valid remark. We adapted the definition of (standardized) mean differences for the comparison between the CTIN+CINP and CINP group according to the comparison between the CTIN+CINP and CTIN group to make this homogenous (Table 3). In addition, we added an additional heading above the pain characteristics and changed the order of the data for the CTIN+CINP group and CINP group to make table 3 clearer and more homogenous for all outcome measures.

3. While the authors do account for multiple comparisons within an outcome measure, they do not do anything to account for the large number of outcomes being assessed.

Thank you for this relevant comment. In our analysis, we focused on controlling for multiple comparisons within each outcome measure. However, by using multivariate analysis of covariance (MANCOVA) on clinical clusters of outcome measures, which were created based on correlated variables, we aimed to reduce the number of independent statistical tests and account for interrelated outcome measures. This resulted in the conversion of 44 variables to 13 clusters of correlated outcome measures and the performance of 13 independent tests.

However, we recognize that the overall number of outcome measures should also be considered when interpreting the results. We encourage readers to interpret the findings in light of the exploratory nature of our study and the potential for chance findings due to multiple comparisons. Therefore we added following sentence to the limitations section: “Sixth, when interpreting the results, the large number of measures should be considered since this may lead to chance findings from multiple comparisons. However, by performing MANCOVA analyses on clusters of correlated outcome measures, we aimed to reduce independent tests and account for interrelated measures. (line 507-510)”

Minor points:

1. lines 149-150: authors might consider rewriting +/- 2 hours and +/- 1 hour and instead give the approximate time it takes to complete the assessments.

Thank you for this remark. We clarified this and changed these sentences into: “After inclusion, participants were invited for a lab visit for experimental assessments (1 h 45 minutes). Prior, participants were asked to fill out an online baseline questionnaire battery (1 hour, fixed order) to acquire information on demographics, in-and exclusion criteria, tinnitus- and pain-related characteristics, psychological and lifestyle factors.”(line 144-145)

2. lines 305-306: "Total score ranges..." is stated in lines 302-303.

Thank you very much for notifying this double report, we removed the sentence “The maximal score is 28 and higher scores indicate more severe insomnia.”(line 297)

3. Authors should clarify the number of participants included in the study. line 350 indicates 48 patients with tinnitus only, while Figure 1 has 47. Numbers in Table 3 also differ from what is stated in line 350 and Figure 1. For example, for Sex, there is information presented for 19 CTIN+CINP, but line 350 says there are only 18. Also, the numbers presented for (dominant) painful side do not seem correct for CINP (and the percentage seems incorrect for Left for CTIN+CINP). Authors should carefully check the numbers in Table 3 and make sure they are consistent within the table as well as with what is presented in Figure 1 and in the text.

Thank you very much for this helpful remark, there were indeed some mistakes in the flowchart and the text for which we apologize. Based on the inclusion questionnaire, 47 CTIN, 19 CTIN + CINP, 29 CINP and 40 HC were found to be eligible and participated to the study (they filled out the online baseline questionnaire and/or participated to the QST measurements). The online baseline questionnaire (prior to the lab visit) was not filled out by one CTIN + CINP and one HC, leading to a total of 47 CTIN, 18 CTIN+CINP, 29 CINP and 39 HC who completed the online baseline questionnaire. For the lab visit (QST measurements), one CTIN and two CTIN + CINP did not reply to the emails for scheduling a lab visit. In addition, 5 CTIN reported a pain score for more than 2/10 on the testing day and were thus excluded. During the measurements a major technical error was present leading to missing data of one CTIN. This resulted in a total of 40 CTIN, 17 CTIN+CINP, 29 CINP and 40 HC of which QST data were collected. During the QST measures, some technical errors occurred, leading to incomplete data for some participants (this is reported in the flowchart (figure 1) by “Incomplete data”), the exact number of participants for each QST measurement is depicted in figure 2A and 2B. We also mentioned additional missing data for specific demographic variables in table 3. To inform the readers about this, we added the sentence: “Additional missing data for a specific variable are listed in the tables or figures.”(line 347-348).

We corrected errors in the population characteristics section (line 345) and the flowchart (figure 1). In addition, we mentioned in table 3 and 5 from which questionnaire(s) (inclusion, baseline or both) presented data were collected and we also added this to the flowchart (figure 1) to make the sample sizes and drop-outs at each stage clearer. Lastly, we corrected the mistake in the numbers and percentages of the (dominant) painful side (table 3). 

Figure 1 revised with changes highlighted in yellow.

4. lines 369-370: Authors refer to Table 5 twice, but I believe this should be Table 3

Thank you for this helpful remark, we indeed referred wrongly to the results displayed in table 5, this was adapted in the text to table 3 (lines 364-365).

5. lines 377: authors refer to differences in the Masseter muscle (PTT) and C5-C6-location (6PPT) between CINP and HC, but these are not significant according to their stated criteria.

Thank you very much for notifying this unfortunate mistake, we corrected this sentence to “The CINP group had significantly lower 6PPTs at the Masseter muscle and Trigeminal nerve.” (lines 371-372)

6. Figures 2 and 3 are described as histograms, but histograms are not actually presented.

Thank you for this considerate remark, we changed the description of the figures to “bar plots” (lines 382 and 421).

7. line 397: change "CTIN and CINP group" to "CTIN and CINP groups"

Thank you for notifying this mistake, we changed "CTIN and CINP group" to "CTIN and CINP groups" (line 392).

8. line 414: authors refer to Fig 3 for the Physical Activity, but this measure is not shown in the figure.

Thank you for notifying this mistake, the referral to figure 3 was removed in the text (line 409).

2. Responses to reviewer 2

1. Please consider rephrasing the third aim in the introduction (ln 112-114) to make it more comprehensible.

Thank you for addressing this unclarity, we changed the last section of the introduction to make the third aim more comprehensible:

“Therefore, the first aim of the present study is to compare signs and symptoms of CS between chronic tinnitus patients with and without chronic idiopathic neck pain (CINP), patients with CINP only and healthy controls. Since psychological and lifestyle factors are shown to be associated with tinnitus (36–38), chronic pain (39) and CS (49–51), and thus can influence the relationship between tinnitus and CS, the second aim is to compare psychological and lifestyle factors between all groups and a third aim is to explore how these signs and symptoms of CS are associated with psychological and lifestyle factors, as well as with factors related to tinnitus and pain. It can be expected that more deterioration in self-reported psychological and lifestyle measures is present in patients with chronic tinnitus or CINP, when compared to HCs and are most prominent in patients with both chronic tinnitus and CINP. It can also be hypothesized that more extensive signs and symptoms of CS are associated with more deterioration in psychological, lifestyle, tinnitus and pain factors.”(line 112-118)

2. Instead of using “/” as a separator in the tables, consider making individual columns.

Thank you for this helpful suggestion. We adapted table 3 and 6 and made individual columns instead of “/” separators to display the frequencies.

3. Page 32, first paragraph, ln 2 from bottom: make sure “and QST measures” should be removed. (the tables are displayed before the discussion, and the pages lost their numerical count on the left handside).

Thank you for notifying this mistake, we removed “and QST measures” in line 490. We also moved the tables after the discussion section and add a continuous line numbering throughout the manuscript.

4. The figure quality and resolution is extremely poor across all figures. The images are blurry and it is impossible to read the numbers and letters on the x and y axis. I suggest the authors upload new figures in better resolution and also increase the font size on the x and y axis.

Thank you for pointing out this problem, we apologize for the inconvenience. We improved the resolution of all figures and also increased the font size of the x-axis and y-axis of all bar plot figures.

---

## [Decision Letter · Decision Letter 1]

2 Aug 2023

Suffering from chronic tinnitus, chronic neck pain, or both: does it impact the presence of signs and symptoms of central sensitization?

PONE-D-23-01457R1

Dear Dr. De Meulemeester,

We’re pleased to inform you that your manuscript has been judged scientifically suitable for publication and will be formally accepted for publication once it meets all outstanding technical requirements.

Kind regards,

Bianka Karshikoff, PhD

Academic Editor

PLOS ONE

Additional Editor Comments (optional):

Reviewers' comments:

Reviewer's Responses to Questions

**Comments to the Author**

1. If the authors have adequately addressed your comments raised in a previous round of review and you feel that this manuscript is now acceptable for publication, you may indicate that here to bypass the “Comments to the Author” section, enter your conflict of interest statement in the “Confidential to Editor” section, and submit your "Accept" recommendation.

Reviewer #1: All comments have been addressed

Reviewer #2: All comments have been addressed

2. Is the manuscript technically sound, and do the data support the conclusions?

Reviewer #1: (No Response)

Reviewer #2: Yes

3. Has the statistical analysis been performed appropriately and rigorously? 

Reviewer #1: (No Response)

Reviewer #2: Yes

4. Have the authors made all data underlying the findings in their manuscript fully available?

Reviewer #1: (No Response)

Reviewer #2: Yes

5. Is the manuscript presented in an intelligible fashion and written in standard English?

Reviewer #1: (No Response)

Reviewer #2: Yes

6. Review Comments to the Author

Reviewer #1: (No Response)

Reviewer #2: The manuscript technically sound with aim, methods and appropriate statistical procedures. The data support the conclusions. The authors state that "The data files will be made publicly available upon publication at www.clinicaltrials.gov (NCT05186259)". The manuscript is written in intelligible standard English. The research meets standards for the ethics of experimentation and research integrity. All comments were addressed by the authors.

7. PLOS authors have the option to publish the peer review history of their article (what does this mean?). If published, this will include your full peer review and any attached files.

Reviewer #1: No

Reviewer #2: No

---

## [Editor Report · Acceptance letter]

15 Aug 2023

PONE-D-23-01457R1 

Suffering from chronic tinnitus, chronic neck pain, or both: does it impact the presence of signs and symptoms of central sensitization? 

Dear Dr. De Meulemeester:

I'm pleased to inform you that your manuscript has been deemed suitable for publication in PLOS ONE. Congratulations! Your manuscript is now with our production department. 

Kind regards, 

on behalf of

Dr. Bianka Karshikoff 

Academic Editor

PLOS ONE